# Reverse Flow Matching: A Unified Framework for Online Reinforcement Learning with Diffusion and Flow Policies

**Zeyang Li** [1]  **Sunbochen Tang** [1]  **Navid Azizan** [1]

## Abstract

Diffusion and flow policies are gaining prominence in online reinforcement learning (RL) due to their expressive power, yet training them efficiently remains a critical challenge. A fundamental difficulty that distinguishes online RL from standard generative modeling is the lack of direct samples from the target Boltzmann distribution defined by the Q-function. To address this, two seemingly distinct families of methods have been proposed for diffusion policies: a noise-expectation family, which uses a weighted average of noise as the training target, and a gradient-expectation family, which employs a weighted average of Q-function gradients. However, it remains unclear how these objectives are formally related, or whether they can be synthesized into a more general formulation. In this paper, we propose a unified framework, reverse flow matching (RFM), which rigorously addresses the problem of training diffusion and flow models without direct target samples. By adopting a reverse inferential perspective, we formulate the training target as a posterior mean estimation problem given an intermediate noisy sample. Crucially, we introduce Langevin Stein operators to construct zero-mean control variates, deriving a general class of estimators that share the same expectation. We show that existing noise-expectation and gradient-expectation methods are simply two specific instances within this broader class. This unified view yields two key advancements: it extends the capability of targeting Boltzmann distributions from diffusion to flow policies, and it enables the principled combination of Q-value and Q-gradient information to form an effective estimator, thereby improving training efficiency

[1]Massachusetts Institute of Technology. Correspondence to: Zeyang Li <zeyang@mit.edu>.

*Proceedings of the 43rd International Conference on Machine Learning*, Seoul, South Korea. PMLR 306, 2026. Copyright 2026 by the author(s).

and stability. We instantiate RFM to train a flow policy in online RL and demonstrate improved performance on continuous-control benchmarks compared to diffusion policy baselines.

## 1. Introduction

Diffusion (Sohl-Dickstein et al., 2015; Ho et al., 2020) and flow (Lipman et al., 2023) have emerged as powerful frameworks for generative modeling, revolutionizing various domains such as image synthesis (Rombach et al., 2022; Esser et al., 2024) and video generation (Ho et al., 2022; Jin et al., 2025). Their ability to model complex, high-dimensional distributions makes them particularly appealing for decision making, where policies often require rich expressiveness to capture multi-modal behaviors in challenging environments. Diffusion and flow policies have demonstrated remarkable success in imitation learning and offline reinforcement learning (RL) (Chi et al., 2025; Ding et al., 2025; Wang et al., 2023; Park et al., 2025), benefiting from the direct availability of expert demonstrations or pre-collected datasets.

However, harnessing diffusion and flow policies in online RL poses significant challenges. A fundamental difficulty distinguishing online RL from standard generative modeling is the lack of direct samples from the target action distribution. In the maximum entropy RL framework (Haarnoja et al., 2018), the improved policy is defined by a Boltzmann distribution over actions, $\pi_{\text{new}}(a \mid s) \propto \exp\left(\frac{1}{\lambda}Q(s, a)\right)$, which is unnormalized and generally intractable to sample from directly. This contrasts sharply with standard generative modeling where training data is readily available. Although some approaches attempt to circumvent this challenge using alternative objectives, they are often hindered by high computational costs, numerical instability, or biased estimators. These limitations lead to suboptimal performance, ultimately restricting the full potential of diffusion and flow policies in online RL. Among methods that do attempt to target the Boltzmann distribution, two seemingly distinct families have been proposed for diffusion policies. The noise-expectation approach (Ma et al., 2025; Dong et al., 2025) constructs training targets via self-normalized importance sampling (SNIS) of noise, utilizing exponentiated

Q-values as weights. The gradient-expectation approach (Akhound-Sadegh et al., 2024; Jain et al., 2025), instead performs SNIS over the gradient of the Q-function. Although both have shown empirical promise, it remains unclear how these objectives relate formally or if they can be synthesized into a more general formulation. Additionally, existing derivations are often rigidly coupled with specific noise schedules (e.g., variance-preserving or variance-exploding), which tends to obscure the underlying principles. Furthermore, these methods have been limited to diffusion policies, leaving the effective training of flow policies to sample from Boltzmann distributions as an open problem.

In this paper, we propose a unified framework, reverse flow matching (RFM), which rigorously addresses the problem of training diffusion and flow models without direct target samples. Our contributions are summarized as follows.

1. We propose a tractable RFM loss that transforms the intractable situation of unavailable target samples into a posterior mean estimation problem.

2. We introduce Langevin Stein operators to construct zero-mean control variates and derive a general class of posterior mean estimators. We show that existing noise-expectation and gradient-expectation methods are two specific instances within this broader class. This framework enables the principled combination of Q-value and Q-gradient information to obtain an effective variance-reduced estimator, thereby improving training efficiency and stability.

3. This unification extends the capability of targeting Boltzmann distributions from diffusion to flow policies. Since flow models admit general source distributions beyond the standard Gaussian, they offer greater flexibility and open the door to incorporating domain knowledge through tailored source choices.

4. We instantiate RFM to train a flow policy in online RL and demonstrate its superior performance on continuous-control benchmarks compared to diffusion policy baselines.

## 2. Related Works

This section discusses prior works on diffusion and flow policies for online RL. We group existing methods into four categories: (1) optimization via differentiable sampling (Wang et al., 2024; Lv et al., 2025; Celik et al., 2025); (2) sampling via Langevin dynamics (Psenka et al., 2024; Ishfaq et al., 2025); (3) iterative weighted regression (Ding et al., 2024; Fan et al., 2025; Ma et al., 2025); and (4) targeting Boltzmann distributions (Akhound-Sadegh et al., 2024; Jain et al., 2025; Ma et al., 2025; Dong et al., 2025). A more detailed discussion is provided in Appendix A.

## 3. Background

### 3.1. Reinforcement Learning

Reinforcement learning (RL) is grounded in the Markov decision process (MDP) framework. An MDP is specified by the tuple $(\mathcal{S}, \mathcal{A}, P, r, d_0, \gamma)$, where $\mathcal{S}$ is the state space, $\mathcal{A}$ is the action space, $P : \mathcal{S} \times \mathcal{A} \to \Delta(\mathcal{S})$ is the transition kernel, $r : \mathcal{S} \times \mathcal{A} \to \mathbb{R}$ is the reward function, $d_0 \in \Delta(\mathcal{S})$ is the initial state distribution, and $\gamma \in (0, 1)$ is the discount factor. $\Delta(\cdot)$ denotes the set of probability distributions over its argument. A policy $\pi : \mathcal{S} \to \Delta(\mathcal{A})$ maps each state to a distribution over actions, and $\pi(a \mid s)$ denotes the probability of taking action $a$ at state $s$ under policy $\pi$. The goal of RL is to find an optimal policy $\pi^*$ that maximizes the expected cumulative discounted reward.

The maximum entropy RL framework augments the standard RL objective with an entropy regularization term to encourage exploration and improve policy robustness. The objective is defined as

$$J(\pi) = \mathbb{E}_{\tau \sim \pi} \left[ \sum_{t=0}^{\infty} \gamma^t \left( r(s_t, a_t) + \lambda \mathcal{H}(\pi(\cdot \mid s_t))) \right) \right],$$

where $\tau$ denotes a trajectory generated by policy $\pi$, $\mathcal{H}(\pi(\cdot \mid s_t)) = -\mathbb{E}_{a_t \sim \pi(\cdot \mid s_t)}[\log \pi(a_t \mid s_t)]$ represents the entropy of policy $\pi$ at state $s_t$, and $\lambda$ is a regularization parameter. The regularized self-consistency operator for a given policy $\pi$ is defined as $(\mathcal{T}_\lambda^\pi Q)(s, a) = r(s, a) + \gamma \mathbb{E}_{s' \sim P(\cdot \mid s, a)} \left[ \mathbb{E}_{a' \sim \pi(\cdot \mid s')} [Q(s', a')] + \lambda \mathcal{H}(\pi(\cdot \mid s')) \right]$, where $Q : \mathcal{S} \times \mathcal{A} \to \mathbb{R}$ denotes the regularized state-action value function, also known as the soft Q-function. The regularized Bellman operator is defined as

$$[\mathcal{T}_\lambda(Q)](s, a) = \max_\pi \left\{ [\mathcal{T}_\lambda^\pi(Q)](s, a) \right\}. \quad (1)$$

The soft Q-function for policy $\pi$ is the fixed point $Q^\pi = \mathcal{T}_\lambda^\pi Q^\pi$, and the optimal soft Q-function satisfies $Q^* = \mathcal{T}_\lambda Q^*$, with corresponding optimal policy $\pi^*$. The soft policy iteration algorithm alternates between soft policy evaluation, which solves for $Q^\pi = \mathcal{T}_\lambda^\pi Q^\pi$ given $\pi$, and soft policy improvement, which updates the policy according to $\pi_{\text{new}} = \underset{\pi}{\operatorname{argmax}} \{\mathcal{T}_\lambda^\pi Q^\pi\}$. The closed-form solution to this optimization problem is a Boltzmann distribution over actions: $\pi_{\text{new}}(a \mid s) \propto \exp\left(\frac{1}{\lambda} Q^\pi(s, a)\right)$.

### 3.2. Diffusion and Flow

Consider a time-indexed family of random variables $\{X_t\}_{t \in [0,1]}$ taking values in $\mathbb{R}^d$, with associated densities $p_t$. In diffusion and flow, the objective is to learn a stochastic or deterministic evolution that transports samples from a source distribution $p_0$ to a target distribution $p_1$. Concretely, the marginal probability path $\{p_t\}_{t \in [0,1]}$ must satisfy the boundary conditions $p_{t=0} = p_0$ and $p_{t=1} = p_1$.

The joint law of the endpoints $(X_0, X_1)$ is denoted by $\nu_{0,1}$, and is referred to as the coupling. The independent coupling $\nu_{0,1}(x_0, x_1) = p_0(x_0)p_1(x_1)$ is a common choice in practice.

The central idea in diffusion and flow models is to construct a conditional probability path $\{p_{t|Z}\}_{t \in [0,1]}$ that interpolates between $p_0$ and $p_1$ given a conditioning variable $Z$. Common choices of $Z$ include the two-sided $Z = (X_0, X_1)$ and the one-sided $Z = X_1$. They lead to the same results, though in different contexts one may be more convenient than the other for ease of presentation. We will use both throughout the paper: $Z$ denotes a generic placeholder, and we specify its form when needed. For brevity, we compress notation of conditioning in subscripts; for example, $p_{t|X_0, X_1}$ as $p_{t|0,1}$.

In flow matching, the evolution of $X_t$ is defined via an ordinary differential equation (ODE)

$$\frac{dX_t}{dt} = v_t(X_t), \quad X_{t=0} \sim p_0, \quad (2)$$

where $v_t(x) : \mathbb{R}^d \to \mathbb{R}^d$ is referred to as the marginal velocity field. Given a conditioning variable $Z$, we denote the conditional velocity field by $v_{t|Z}(\cdot \mid Z)$. $v_t(x)$ is the conditional expectation of $v_{t|Z}$ given $X_t = x$:

$$\begin{aligned}
v_t(x) &= \mathbb{E}\left[ v_{t|0,1}\left( X_t \mid X_0, X_1 \right) \mid X_t = x \right] \\
&= \mathbb{E}\left[ v_{t|1}\left( X_t \mid X_1 \right) \mid X_t = x \right].
\end{aligned} \quad (3)$$

To learn a parameterized velocity field $v_t^\theta(x)$, the (conceptual) flow matching loss regresses toward the marginal velocity field $v_t(x)$:

$$\mathcal{L}_{\text{FM}}(\theta) = \mathbb{E}_{t \sim \mathcal{U}[0,1], X_t \sim p_t}\left[ \left\| v_t^\theta(X_t) - v_t(X_t) \right\|_2^2 \right], \quad (4)$$

which is generally intractable since $v_t(x)$ typically admits no closed-form expression. The conditional flow matching loss instead regresses $v_t^\theta$ onto the conditional velocity field $v_{t|Z}$:

$$\begin{aligned}
\mathcal{L}_{\text{CFM}}(\theta) &= \mathbb{E}_{\substack{t \sim \mathcal{U}[0,1], \\ (X_0, X_1) \sim \nu_{0,1}, \\ X_t \sim p_{t|0,1}}}\left[ \left\| v_t^\theta(X_t) - v_{t|0,1}(X_t \mid X_0, X_1) \right\|_2^2 \right] \\
&= \mathbb{E}_{\substack{t \sim \mathcal{U}[0,1], \\ X_1 \sim p_1, X_t \sim p_{t|1}}}\left[ \left\| v_t^\theta(X_t) - v_{t|1}(X_t \mid X_1) \right\|_2^2 \right].
\end{aligned} \quad (5)$$

We recall the following equivalence between the marginal and conditional objectives (Lipman et al., 2023).

**Lemma 3.1.** *Under mild regularity conditions, $\mathcal{L}_{\text{FM}}$ and $\mathcal{L}_{\text{CFM}}$ share the same set of global minimizers. Moreover, their gradients with respect to $\theta$ coincide, i.e., $\nabla_\theta \mathcal{L}_{\text{FM}}(\theta) = \nabla_\theta \mathcal{L}_{\text{CFM}}(\theta)$ for all $\theta$.*

A widely adopted choice to construct a conditional probability path is via the linear interpolation $X_t = \alpha_t X_1 + \beta_t X_0$, where the schedule $(\alpha_t, \beta_t)$ satisfies the boundary conditions $\alpha_0 = 0$, $\alpha_1 = 1$, $\beta_0 = 1$, and $\beta_1 = 0$. Under this construction, for two-sided conditioning, $p_{t|0,1}(x_t \mid x_0, x_1) = \delta(x_t - (\alpha_t x_1 + \beta_t x_0))$ is a Dirac measure concentrated on the interpolant, and the corresponding conditional velocity field is $v_{t|0,1}(X_t \mid X_0, X_1) = \dot{\alpha}_t X_1 + \dot{\beta}_t X_0$. For one-sided conditioning, we have $v_{t|1}(X_t \mid X_1) = \dot{\alpha}_t X_1 + \dot{\beta}_t \frac{X_t - \alpha_t X_1}{\beta_t}$.

The linear interpolation $X_t = \alpha_t X_1 + \beta_t X_0$ encompasses many popular diffusion and flow models as special cases, up to particular choices of the schedule $(\alpha_t, \beta_t)$, time conventions, and prediction parameterization. Moreover, we have $v_t(X_t) = \dot{\alpha}_t \mathbb{E}[X_1 \mid X_t] + \dot{\beta}_t \mathbb{E}[X_0 \mid X_t]$, and, taking conditional expectations in the interpolation itself, $X_t = \alpha_t \mathbb{E}[X_1 \mid X_t] + \beta_t \mathbb{E}[X_0 \mid X_t]$. Thus, the data-prediction and noise-prediction parameterizations can be recovered by solving these equations for $\mathbb{E}[X_1 \mid X_t]$ or $\mathbb{E}[X_0 \mid X_t]$ respectively.

### 3.3. Control Variates

Control variates are a classical technique for reducing the variance of Monte Carlo estimators. Suppose we need to estimate an expectation $\mu = \mathbb{E}_{X \sim q}[f(X)]$ for a measurable function $f : \mathbb{R}^d \to \mathbb{R}$, using samples from $q$. Let $h : \mathbb{R}^d \to \mathbb{R}$ be an auxiliary function whose mean $\mathbb{E}_q[h(X)]$ is known. Then, for any coefficient $\eta \in \mathbb{R}$,

$$\mathbb{E}_{X \sim q}[f(X)] = \mathbb{E}_{X \sim q}\left[ f(X) + \eta\big(h(X) - \mathbb{E}_q[h(X)]\big) \right].$$

Hence we can replace $f(X)$ by $f(X) + \eta\big(h(X) - \mathbb{E}_q[h(X)]\big)$ in Monte Carlo estimation, and choose $h$ and $\eta$ wisely to reduce variance.

## 4. Reverse Flow Matching

Training the velocity field $v_t^\theta$ with the conditional flow matching loss requires samples from the target distribution $p_1$. As indicated by (5), this does not require an explicit form for $p_1$, only the ability to draw samples from it. In some applications, however, the situation is reversed: $p_1$ is known (often only up to a normalizing constant), but an efficient sampler is unavailable. A canonical example is sampling from Boltzmann distributions, which appear widely across scientific domains, and, in our problem of interest, reinforcement learning (RL).

Suppose we parameterize the policy $\pi^\theta(a \mid s)$ using a diffusion or flow model. The soft policy improvement step requires updating the policy towards the Boltzmann distribution $\pi_{\text{new}}(a \mid s) \propto \exp\left(\frac{1}{\lambda}Q(s,a)\right)$. The core challenge is that we only have access to $Q$ and cannot efficiently generate samples from $\pi_{\text{new}}$ to train the policy via condi-

tional flow matching loss. An alternative method is to drop the closed-form expression of $\pi_{\text{new}}$ and return to the underlying optimization problem $\pi_{\text{new}} = \operatorname{argmax}_\pi \{\mathcal{T}_\lambda^\pi Q^\pi\} = \operatorname{argmax}_\pi \{\mathbb{E}_{a \sim \pi(\cdot|s)}[Q(s,a)] + \lambda \mathcal{H}(\pi(\cdot \mid s))\}$. Then the parameters of $\pi^\theta$ can be optimized directly via gradient ascent. However, for diffusion or flow policies this approach is fundamentally flawed: it requires backpropagating through the entire sampling procedure, which is computationally expensive and numerically unstable.

In this section, we derive a reverse flow matching loss that sidesteps the requirement for direct samples from $p_1$, relying instead on posterior mean estimation. Furthermore, we introduce Langevin Stein operators to construct zero-mean control variates, yielding a general class of estimators that share the same expectation. For simplicity, we specialize to the independent coupling $\nu_{0,1}(x_0, x_1) = p_0(x_0)p_1(x_1)$ and the linear interpolation $X_t = \alpha_t X_1 + \beta_t X_0$, but the derivations apply to more general settings. Due to space constraints, we defer proofs of all theoretical results to Appendix C.

### 4.1. From Forward Construction to Reverse Inference

Standard conditional flow matching operates on a forward, constructive principle: we sample the source noise $X_0 \sim p_0$ and target data $X_1 \sim p_1$, then synthesize the intermediate state through the prescribed interpolation $X_t = \alpha_t X_1 + \beta_t X_0$. Training effectively becomes a supervised regression problem conditioned on these known endpoints. This workflow, however, relies entirely on the ability to efficiently sample pairs $(X_0, X_1)$.

In our setting, direct samples from $p_1$ are unavailable, and $p_1$ is only known up to a normalizing constant. This breaks the forward pipeline and necessitates a shift to a reverse, inferential viewpoint. Instead of manufacturing $X_t$ from known components, we treat $X_t$ as observed evidence and $X_0$ as a latent variable that explains its origin. Since the interpolation is a rigid constraint, any hypothesized noise value $x_0$ together with the observation $x_t$ uniquely determines the implied target endpoint $x_1(x_0, x_t) = \frac{1}{\alpha_t}x_t - \frac{\beta_t}{\alpha_t}x_0$. Hence a candidate noise sample $x_0$ is plausible if it is likely under the prior $p_0$ and implies a target $x_1(x_0, x_t)$ that is likely under the target distribution $p_1$. This intuition is formalized by Bayes' theorem, which yields the posterior distribution of the noise $X_0$ given $X_t$:

$$q_{0|t}^*(x_0 \mid x_t) \propto p_0(x_0)\, p_1\left(\frac{1}{\alpha_t}x_t - \frac{\beta_t}{\alpha_t}x_0\right). \quad (6)$$

Crucially, this logic applies symmetrically. The interpolation $X_t = \alpha_t X_1 + \beta_t X_0$ restricts the joint distribution of $(X_0, X_1)$ to a linear manifold passing through $X_t$. Consequently, inferring the noise $X_0$ is mathematically equivalent

to inferring the data $X_1$. This leads to data posterior

$$q_{1|t}^*(x_1 \mid x_t) \propto p_1(x_1)\, p_0\left(\frac{1}{\beta_t}x_t - \frac{\alpha_t}{\beta_t}x_1\right). \quad (7)$$

Both $q_{0|t}^*$ and $q_{1|t}^*$ represent marginal perspectives of the same underlying joint posterior coupling $q_{0,1|t}^*(x_0, x_1 \mid x_t) \propto p_0(x_0)\, p_1(x_1)\, \delta\left(x_t - (\alpha_t x_1 + \beta_t x_0)\right)$.

We can now introduce the reverse flow matching loss. In contrast to conditional flow matching, we replace the unavailable forward samples with ones drawn from the posterior distributions:

$$\mathcal{L}_{\text{RFM}}(\theta) = \mathbb{E}_{\substack{t \sim \mathcal{U}[0,1],\, X_t \sim \hat{p}_t, \\ (X_0, X_1) \sim q_{0,1|t}^*}} \left[\left\| v_t^\theta(X_t) - (\dot{\alpha}_t X_1 + \dot{\beta}_t X_0) \right\|_2^2\right]$$

$$= \mathbb{E}_{\substack{t \sim \mathcal{U}[0,1],\, X_t \sim \hat{p}_t, \\ X_0 \sim q_{0|t}^*, \\ X_1 \sim \delta\left(\frac{1}{\alpha_t}X_t - \frac{\beta_t}{\alpha_t}X_0\right)}} \left[\left\| v_t^\theta(X_t) - (\dot{\alpha}_t X_1 + \dot{\beta}_t X_0) \right\|_2^2\right]$$

$$= \mathbb{E}_{\substack{t \sim \mathcal{U}[0,1],\, X_t \sim \hat{p}_t, \\ X_1 \sim q_{1|t}^*, \\ X_0 \sim \delta\left(\frac{1}{\beta_t}X_t - \frac{\alpha_t}{\beta_t}X_1\right)}} \left[\left\| v_t^\theta(X_t) - (\dot{\alpha}_t X_1 + \dot{\beta}_t X_0) \right\|_2^2\right],$$

$$(8)$$

where $\hat{p}_t$ is a proposal distribution we choose to sample $X_t$ from. Note that $\hat{p}_t$ is not the marginal density $p_t$ in the flow matching loss (4), since we do not have access to it. The first line of (8) shows the general form, while the second and third lines represent implementations using noise-posterior and data-posterior sampling, respectively.

Furthermore, we can push the expectation over $X_0$ or $X_1$ inside the squared norm, yielding simpler objectives. The noise-posterior form is defined as

$$\mathcal{L}_{\text{RFM-N}}(\theta) = \mathbb{E}_{t \sim \mathcal{U}[0,1], X_t \sim \hat{p}_t}$$
$$\left[\left\| v_t^\theta(X_t) - \left(\frac{\dot{\alpha}_t}{\alpha_t}X_t + \frac{\alpha_t \dot{\beta}_t - \dot{\alpha}_t \beta_t}{\alpha_t}\mathbb{E}_{X_0 \sim q_{0|t}^*}[X_0]\right) \right\|_2^2\right], \quad (9)$$

and the data-posterior form is defined as

$$\mathcal{L}_{\text{RFM-D}}(\theta) = \mathbb{E}_{t \sim \mathcal{U}[0,1], X_t \sim \hat{p}_t}$$
$$\left[\left\| v_t^\theta(X_t) - \left(\frac{\dot{\beta}_t}{\beta_t}X_t + \frac{\beta_t \dot{\alpha}_t - \dot{\beta}_t \alpha_t}{\beta_t}\mathbb{E}_{X_1 \sim q_{1|t}^*}[X_1]\right) \right\|_2^2\right]. \quad (10)$$

We state the equivalence of these objectives in the following proposition.

**Proposition 4.1.** *The objectives $\mathcal{L}_{\text{RFM}}(\theta)$, $\mathcal{L}_{\text{RFM-N}}(\theta)$, and $\mathcal{L}_{\text{RFM-D}}(\theta)$ differ only by additive constants independent of $\theta$. Therefore, they share the same set of global minimizers and have identical gradients with respect to $\theta$.*

We next establish the equivalence between reverse flow matching and conditional flow matching.

**Theorem 4.2.** *Assume that for almost every $t \in [0,1]$, the true marginal distribution $p_t$ (used in conditional flow*

*matching) and the proposal distribution $\hat{p}_t$ (used in reverse flow matching) are mutually absolutely continuous. Assume that the parameterized function class $\{v_t^\theta : \theta \in \Theta\}$ is sufficiently rich such that the regression objectives attain their global minima. Then the objectives $\mathcal{L}_{\mathrm{RFM\text{-}N}}$, $\mathcal{L}_{\mathrm{RFM\text{-}D}}$, and $\mathcal{L}_{\mathrm{CFM}}$ share the same set of global minimizers.*

Note that although the global minimizers coincide, the resulting optimization dynamics may differ. In particular, the gradient of the reverse flow matching objectives depends on the choice of proposal $\hat{p}_t$. Consequently, $\nabla_\theta \mathcal{L}_{\mathrm{RFM}}(\theta)$ generally differs from $\nabla_\theta \mathcal{L}_{\mathrm{CFM}}(\theta)$, except in the special case where $\hat{p}_t$ matches the true marginal $p_t$ in (5).

*Remark* 4.3. The reverse flow matching framework flexibly accommodates various network parameterizations. While (9) and (10) are formulated for velocity prediction, the training target can be readily adapted: for data prediction, one can regress directly onto $\mathbb{E}_{X_1 \sim q_{1|t}^*(\cdot|X_t)}[X_1]$, and for noise prediction, onto $\mathbb{E}_{X_0 \sim q_{0|t}^*(\cdot|X_t)}[X_0]$. For score prediction with a Gaussian source $p_0$, the target becomes $-\frac{1}{\beta_t}\mathbb{E}_{X_0 \sim q_{0|t}^*(\cdot|X_t)}[X_0]$. The framework naturally subsumes standard diffusion models through appropriate choices of the schedule $(\alpha_t, \beta_t)$. For instance, the variance-exploding (VE) schedule is recovered by setting $\alpha_t = 1$ and $\beta_t = \sigma_{\min}\left(\frac{\sigma_{\max}}{\sigma_{\min}}\right)^{1-t}$ (forward-time convention). Furthermore, we show in Appendix D that the proposed method can be applied to train score-based models even when the source distribution $p_0$ is non-Gaussian, for which we term reverse score matching.

## 4.2. Langevin Stein Operators

The reverse flow matching approach provides a principled framework for training diffusion and flow models when only an unnormalized target density is available. From (9) and (10), the key computational challenge is estimating the posterior means $\mathbb{E}_{X_0 \sim q_{0|t}^*(\cdot|x_t)}[X_0]$ or $\mathbb{E}_{X_1 \sim q_{1|t}^*(\cdot|x_t)}[X_1]$ for a given $x_t$. We focus here on the noise posterior. The results apply symmetrically to the data posterior.

Recall that $q_{0|t}^*(x_0 \mid x_t) \propto p_0(x_0)\, p_1\left(\frac{1}{\alpha_t}x_t - \frac{\beta_t}{\alpha_t}x_0\right)$ is known only up to a normalizing constant. A standard approach to estimate $\mathbb{E}_{X_0 \sim q_{0|t}^*(\cdot|x_t)}[X_0]$ in this setting is self-normalized importance sampling (SNIS). Given $K$ samples $\{X_0^{(i)}\}_{i=1}^K$ drawn from a chosen proposal distribution $\bar{q}$, the SNIS estimator is

$$\hat{\mu}_{\mathrm{SNIS}}\left[X_0 \mid t, x_t\right] = \frac{\sum_{i=1}^K w(X_0^{(i)}, x_t) X_0^{(i)}}{\sum_{i=1}^K w(X_0^{(i)}, x_t)}, \qquad (11)$$

where the unnormalized importance weights are $w(x_0, x_t) = \frac{p_0(x_0)\tilde{p}_1\left(\frac{1}{\alpha_t}x_t - \frac{\beta_t}{\alpha_t}x_0\right)}{\bar{q}(x_0)}$. $\tilde{p}_1$ denotes the unnormalized part of $p_1$. A simple choice is $\bar{q} = p_0$, in which case

the weights simplify to $w(x_0, x_t) = \tilde{p}_1\left(\frac{1}{\alpha_t}x_t - \frac{\beta_t}{\alpha_t}x_0\right)$.

From a practical standpoint, the variance of the posterior mean estimator is crucial. Under a fixed computational budget (i.e., a fixed number of samples $K$), a lower-variance estimator leads to more reliable estimates and therefore more stable training. To this end, we introduce Langevin Stein operators (Oates et al., 2017; Gorham & Mackey, 2017) and leverage them to construct control variates that effectively reduce the variance of the posterior mean estimation.

**Definition 4.4** (Langevin Stein operator (Oates et al., 2017; Gorham & Mackey, 2017)). Let $p$ be a continuously differentiable density on $\mathbb{R}^d$, and let $\phi : \mathbb{R}^d \to \mathbb{R}^d$ be a continuously differentiable vector field. The Langevin Stein operator $\mathcal{T}_p$, associated with $p$, acts on $\phi$ as

$$(\mathcal{T}_p\phi)(x) = \nabla \cdot \phi(x) + \phi(x) \cdot \nabla \log p(x),$$

where $\nabla \cdot \phi(x) = \sum_{i=1}^d \frac{\partial \phi_i(x)}{\partial x_i}$ is the divergence. In particular, $\mathcal{T}_p\phi : \mathbb{R}^d \to \mathbb{R}$ is scalar-valued.

**Lemma 4.5.** *Let $p$ be a continuously differentiable density on $\mathbb{R}^d$, and let $\phi : \mathbb{R}^d \to \mathbb{R}^d$ be continuously differentiable. Assume $\mathbb{E}_{X \sim p}[|(\mathcal{T}_p\phi)(X)|] < \infty$ and that*

$$\lim_{R \to \infty} \int_{\partial B_R} p(x)\, \phi(x) \cdot n(x)\, dS(x) = 0,$$

*where $B_R := \{x : \|x\|_2 \leq R\}$ and $n(x)$ denotes the outward unit normal on $\partial B_R$. Then*

$$\mathbb{E}_{X \sim p}[(\mathcal{T}_p\phi)(X)] = 0.$$

Lemma 4.5 provides a general recipe for constructing scalar zero-mean control variates under a suitably regular density $p$. In our setting, however, the posterior mean is vector-valued in $\mathbb{R}^d$. We therefore extend the Langevin Stein construction to produce vector-valued zero-mean control variates via matrix-valued test functions.

**Definition 4.6** (generalized Langevin Stein operator). Let $p$ be a continuously differentiable density on $\mathbb{R}^d$. Let $\Phi : \mathbb{R}^d \to \mathbb{R}^{d \times m}$ be continuously differentiable, and write $\Phi = [\phi_1, \cdots, \phi_m]$ where each column $\phi_j : \mathbb{R}^d \to \mathbb{R}^d$ is a vector field. Define the generalized Langevin Stein operator $\mathcal{T}_{p,m}$ by

$$(\mathcal{T}_{p,m}\Phi)(x) := \begin{pmatrix} (\mathcal{T}_p\phi_1)(x) \\ \vdots \\ (\mathcal{T}_p\phi_m)(x) \end{pmatrix}$$
$$= \nabla \cdot \Phi(x) + \Phi(x)^\top \nabla \log p(x)$$

where $\nabla \cdot \Phi(x) \in \mathbb{R}^m$ denotes the column-wise divergence with components

$$(\nabla \cdot \Phi(x))_j = \sum_{i=1}^d \frac{\partial \Phi_{ij}(x)}{\partial x_i},$$

for $j = 1, \cdots, m$. In particular, $\mathcal{T}_{p,m}\Phi : \mathbb{R}^d \to \mathbb{R}^m$ is vector-valued.

**Proposition 4.7.** *Assume that each column $\phi_j$ of $\Phi$ satisfies the integrability and boundary conditions of Lemma 4.5. Then*

$$\mathbb{E}_{X \sim p}[(\mathcal{T}_{p,m}\Phi)(X)] = 0 \in \mathbb{R}^m.$$

According to Proposition 4.7, setting $m = d$ allows us to use any (suitably regular) matrix-valued test function $\Phi_t(\cdot, x_t) : \mathbb{R}^d \to \mathbb{R}^{d \times d}$ to construct a vector-valued zero-mean control variate under the posterior density $q^*_{0|t}(\cdot \mid x_t)$. We denote the test function as $\Phi_t(\cdot, x_t)$ to highlight its dependence on $t$ and $x_t$, which act as fixed parameters. Define

$$g_{\Phi_t}(x_0, x_t) := (\mathcal{T}_{q^*_{0|t}(\cdot|x_t),d}\Phi_t(\cdot, x_t))(x_0) \in \mathbb{R}^d,$$

It follows that

$$\mathbb{E}_{X_0 \sim q^*_{0|t}(\cdot|x_t)}[g_{\Phi_t}(X_0, x_t)] = 0 \in \mathbb{R}^d.$$

Consequently, $g_{\Phi_t}(X_0, x_t)$ is a vector-valued zero-mean control variate under $q^*_{0|t}(\cdot \mid x_t)$, and adding $g_{\Phi_t}$ preserves the posterior mean:

$$\mathbb{E}_{X_0 \sim q^*_{0|t}(\cdot|x_t)}[X_0] = \mathbb{E}_{X_0 \sim q^*_{0|t}(\cdot|x_t)}[X_0 + g_{\Phi_t}(X_0, x_t)].$$

The remaining question is how to choose $\Phi_t(\cdot, x_t)$. Theoretically, under suitable regularity assumptions on $q^*_{0|t}(\cdot \mid x_t)$ and the admissible class of test functions, there may exist an optimal choice $\Phi_t^\star$ that eliminates the variance entirely. This zero-variance condition is characterized by the following proposition.

**Proposition 4.8.** *Let $\mu_{0|t}(x_t) = \mathbb{E}_{X_0 \sim q^*_{0|t}(\cdot|x_t)}[X_0]$. Assume $\Phi_t(\cdot, x_t)$ is admissible so that Proposition 4.7 applies. Then the estimator*

$$X_0 + (\mathcal{T}_{q^*_{0|t}(\cdot|x_t),d}\Phi_t(\cdot, x_t))(X_0)$$

*has zero variance under $q^*_{0|t}(\cdot \mid x_t)$ if and only if*

$$\left(\mathcal{T}_{q^*_{0|t}(\cdot|x_t),d}\Phi_t(\cdot, x_t)\right)(x_0) = \mu_{0|t}(x_t) - x_0 \qquad (12)$$

*holds on the support of $q^*_{0|t}(\cdot \mid x_t)$.*

Solving the functional equation (12) is generally intractable. Nevertheless, it motivates minimizing the estimator variance over a parametric family $\{\Phi_\psi : \psi \in \Psi\}$, where $\Psi$ is an appropriate parameter space.

As a concrete example, consider the class of diagonal test functions

$$\Phi_t(x_0, x_t) = \mathrm{diag}\{h_{t,1}(x_0, x_t), \cdots, h_{t,d}(x_0, x_t)\},$$

where each $h_{t,j}(\cdot, x_t) : \mathbb{R}^d \to \mathbb{R}$ is a scalar function for $j = 1, \cdots, d$. In this case, the induced control variate takes the coordinate-wise form

$$g_{\Phi_t}(x_0, x_t) =$$
$$\begin{pmatrix} \partial_{x_{0,1}} h_{t,1}(x_0, x_t) + h_{t,1}(x_0, x_t) \partial_{x_{0,1}} \log q^*_{0|t}(x_0 \mid x_t) \\ \partial_{x_{0,2}} h_{t,2}(x_0, x_t) + h_{t,2}(x_0, x_t) \partial_{x_{0,2}} \log q^*_{0|t}(x_0 \mid x_t) \\ \vdots \\ \partial_{x_{0,d}} h_{t,d}(x_0, x_t) + h_{t,d}(x_0, x_t) \partial_{x_{0,d}} \log q^*_{0|t}(x_0 \mid x_t) \end{pmatrix},$$
$$(13)$$

where $\partial_{x_{0,j}}$ denotes the partial derivative with respect to the $j$-th coordinate of $x_0$.

Denote the posterior score function with respect to $x_0$ by

$$s^*_{0|t}(x_0, x_t) := \nabla_{x_0} \log q^*_{0|t}(x_0 \mid x_t) \in \mathbb{R}^d.$$

If we further restrict $h_{t,j}(x_0, x_t) \equiv \Lambda_j$ (i.e., constant function) for each $j = 1, \cdots, d$, then $\partial_{x_{0,j}} h_{t,j}(x_0, x_t) = 0$ and (13) simplifies to

$$g_{\Phi_t}(x_0, x_t) = \mathrm{diag}(\Lambda) s^*_{0|t}(x_0, x_t), \qquad (14)$$

where $\Lambda = (\Lambda_1, \cdots, \Lambda_d)^\top \in \mathbb{R}^d$. By construction, $\mathbb{E}_{X_0 \sim q^*_{0|t}(\cdot|x_t)}[g_{\Phi_t}(X_0, x_t)] = 0$. Therefore,

$$\mathbb{E}_{X_0 \sim q^*_{0|t}}[X_0] = \mathbb{E}_{X_0 \sim q^*_{0|t}}\left[X_0 + \mathrm{diag}(\Lambda) s^*_{0|t}(X_0, x_t)\right].$$

Accordingly, given samples $\{X_0^{(i)}\}_{i=1}^K$ from a proposal distribution $\bar{q}$, with corresponding unnormalized importance weights $w(X_0^{(i)}, x_t)$, we define the SNIS estimator with the control variate

$$\hat{\mu}_{\text{SNIS-CV}}[X_0 \mid t, x_t; \Lambda] =$$
$$\frac{\sum_{i=1}^K w(X_0^{(i)}, x_t)\left(X_0^{(i)} + \mathrm{diag}(\Lambda) s^*_{0|t}(X_0^{(i)}, x_t)\right)}{\sum_{i=1}^K w(X_0^{(i)}, x_t)}.$$
$$(15)$$

To minimize the variance of SNIS estimator (15), we have the following result.

**Proposition 4.9.** *Fix $t$ and $x_t$. Let $X_0 \sim \bar{q}$ and let $w(X_0, x_t)$ denote the corresponding unnormalized importance weight. Recall the notations $s^*_{0|t,j}(x_0, x_t) := \partial_{x_{0,j}} \log q^*_{0|t}(x_0 \mid x_t)$ and $\mu_{0|t}(x_t) = \mathbb{E}_{X_0 \sim q^*_{0|t}(\cdot|x_t)}[X_0]$. Assume $\mathbb{E}_{\bar{q}}[w(X_0)] < \infty$ and $\mathbb{E}_{\bar{q}}[w(X_0)^2 \|f_\Lambda(X_0, x_t)\|_2^2] < \infty$. Among all constant diagonal choices $h_{t,j} \equiv \Lambda_j$, the coefficients that minimize the asymptotic variance of estimator (15) are given componentwise by*

$$\Lambda_j^* = -\frac{\mathbb{E}_{\bar{q}}\left[w(X_0, x_t)^2\left(X_{0,j} - \mu_{0|t,j}(x_t)\right) s^*_{0|t,j}(X_0, x_t)\right]}{\mathbb{E}_{\bar{q}}\left[w(X_0, x_t)^2\left(s^*_{0|t,j}(X_0, x_t)\right)^2\right]},$$
$$(16)$$

*for $j = 1, \cdots, d$.*

A special case of interest is the isotropic restriction $\Lambda_1 = \cdots = \Lambda_d = \eta$ for some scalar $\eta \in \mathbb{R}$. We have the following result.

**Proposition 4.10.** *Take the same assumptions as in Proposition 4.9. If further apply the isotropic restriction $\Lambda_1 = \cdots = \Lambda_d = \eta$ (equivalently, $\Phi_t = \eta I_d$), then the coefficient that minimizes the asymptotic variance of the corresponding SNIS estimator is*

$$\eta^* = -\frac{\mathbb{E}_{\bar{q}}\Big[w(X_0, x_t)^2 \big(X_0 - \mu_{0|t}(x_t)\big)^\top s^*_{0|t}(X_0, x_t)\Big]}{\mathbb{E}_{\bar{q}}\Big[w(X_0, x_t)^2 \|s^*_{0|t}(X_0, x_t)\|_2^2\Big]}. \tag{17}$$

*Remark* 4.11. Propositions 4.9 and 4.10 exemplify the strategy of parameterizing the matrix test function $\Phi_t$ and minimizing the variance over the resulting parameter space. More expressive parametrizations are also possible. For example, we may let the entries of $\Phi_t$ depend on $x_0$ through polynomials, or use richer feature expansions. We can also parameterize $\Phi_t(x_0, x_t)$ with a neural network and train it from samples to amortize variance reduction across $(t, x_t)$, though special care is needed to enforce the regularity conditions.

*Remark* 4.12. We focus on the SNIS approach for estimating the posterior means $\mathbb{E}_{X_0 \sim q^*_{0|t}(\cdot|x_t)}[X_0]$ and $\mathbb{E}_{X_1 \sim q^*_{1|t}(\cdot|x_t)}[X_1]$, due to its simplicity and seamless integration into the RL training loop. However, the scope of our contribution extends beyond this specific estimator. The proposed control variates from Langevin Stein operators are broadly applicable to advanced estimation methods, such as Markov chain Monte Carlo (MCMC) and sequential Monte Carlo (SMC).

*Remark* 4.13. While this paper centers on training diffusion and flow models via the reverse flow matching framework, the posterior means $\mathbb{E}_{X_0 \sim q^*_{0|t}(\cdot|x_t)}[X_0]$ and $\mathbb{E}_{X_1 \sim q^*_{1|t}(\cdot|x_t)}[X_1]$ used as supervision signals can also be directly employed for training-free sampling. Given $t$ and $x_t$, we can estimate the posterior mean to obtain the corresponding velocity (or score), enabling integration of the ODE (or SDE) over $t \in [0, 1]$. Moreover, the construction of control variates via Langevin Stein operators can be applied in this setting to reduce estimation variance, thereby improving the quality of generated samples. Existing works that may benefit from our control variate constructions include (Pan et al., 2024; Huang et al., 2024; Grenioux et al., 2024).

### 4.3. Application to Boltzmann Distributions

Suppose the data distribution is of the Boltzmann form $p_1(x_1) \propto \exp\big(\frac{1}{\lambda} Q(x_1)\big)$. Applying the control variate constructions to this specific $p_1$, we have the following result.

**Theorem 4.14.** *Assume the target density has the Boltzmann form $p_1(x_1) \propto \exp\big(\frac{1}{\lambda} Q(x_1)\big)$. Let $\Phi_t(x_0, x_t) =$* $\mathrm{diag}\{h_{t,1}(x_0, x_t), \ldots, h_{t,d}(x_0, x_t)\}$ *be a diagonal test function with constant entries $h_{t,j}(x_0, x_t) \equiv \Lambda_j$, and write $\Lambda = (\Lambda_1, \ldots, \Lambda_d)^\top$. Then the induced control variate satisfies*

$$\begin{aligned} g_{\Phi_t}(x_0, x_t) &= \mathrm{diag}(\Lambda) \, \nabla_{x_0} \log p_0(x_0) \\ &\quad - \frac{1}{\lambda} \frac{\beta_t}{\alpha_t} \mathrm{diag}(\Lambda) \left[\nabla_{x_1} Q(x_1)\right]_{x_1 = \frac{1}{\alpha_t} x_t - \frac{\beta_t}{\alpha_t} x_0}, \end{aligned} \tag{18}$$

*and the posterior mean estimator can be expressed as*

$$\begin{aligned} \mu_{0|t}(x_t) &= \mathbb{E}_{X_0 \sim q^*_{0|t}(\cdot|x_t)}\Bigg[X_0 + \mathrm{diag}(\Lambda) \, \nabla_{x_0} \log p_0(X_0) \\ &\quad - \frac{1}{\lambda} \frac{\beta_t}{\alpha_t} \, \mathrm{diag}(\Lambda) \left[\nabla_{x_1} Q(x_1)\right]_{x_1 = \frac{1}{\alpha_t} x_t - \frac{\beta_t}{\alpha_t} X_0}\Bigg]. \end{aligned} \tag{19}$$

*Moreover, if $p_0 = \mathcal{N}(0, I_d)$, we additionally have the identity*

$$\begin{aligned} \mathbb{E}_{X_0 \sim q^*_{0|t}(\cdot|x_t)}[X_0] &= \mathbb{E}_{X_0 \sim q^*_{0|t}(\cdot|x_t)} \\ &\quad \left[-\frac{1}{\lambda} \frac{\beta_t}{\alpha_t} \left[\nabla_{x_1} Q(x_1)\right]_{x_1 = \frac{1}{\alpha_t} x_t - \frac{\beta_t}{\alpha_t} X_0}\right]. \end{aligned} \tag{20}$$

*If we further impose isotropic coefficients $\Lambda_j \equiv \eta$ for $j = 1, \ldots, d$ (equivalently, $\Phi_t = \eta I_d$), then the posterior mean simplifies to a linear combination:*

$$\begin{aligned} \mu_{0|t}(x_t) &= (1 - \eta) \, \mathbb{E}_{X_0 \sim q^*_{0|t}(\cdot|x_t)}[X_0] \\ &\quad + \eta \, \mathbb{E}_{X_0 \sim q^*_{0|t}(\cdot|x_t)}\left[-\frac{1}{\lambda} \frac{\beta_t}{\alpha_t} \left[\nabla_{x_1} Q(x_1)\right]_{x_1 = \frac{1}{\alpha_t} x_t - \frac{\beta_t}{\alpha_t} X_0}\right]. \end{aligned} \tag{21}$$

Theorem 4.14 characterizes a general family of posterior mean estimators for Boltzmann target distributions, whose variance can be controlled through the choice of $\Lambda$ and $\eta$ in (19) and (21). As shown in Propositions 4.9 and 4.10, these coefficients admit variance-minimizing choices, yielding SNIS estimators with reduced variance and, consequently, more stable and accurate supervision signals for training diffusion and flow models. Importantly, our framework also provides a unifying view of existing approaches for targeting Boltzmann distributions in online RL (Akhound-Sadegh et al., 2024; Jain et al., 2025; Ma et al., 2025; Dong et al., 2025). In particular, the noise-expectation and gradient-expectation families arise as two special cases of our general formulation by setting $\eta = 0$ and $\eta = 1$, respectively. A detailed mapping from these prior methods to our formulation is provided in Appendix B.

### 4.4. Online Reinforcement Learning with Flow Policies

We now instantiate reverse flow matching for flow policies in online RL. The extension to diffusion policies is straightforward.

We follow the actor-critic framework. We adopt the double Q-network design with two critics $Q^{\omega_1}(s, a)$ and $Q^{\omega_2}(s, a)$. Given a set $\mathcal{D}$ of collected transitions, the critic losses are given by $\mathcal{L}_Q(\omega_i) = \mathbb{E}_{(s,a,r,s')\sim\mathcal{D}}\left[\left(Q^{\omega_i}(s, a) - \hat{Q}\right)^2\right]$, for $i \in \{1, 2\}$, where $\bar{\omega}_i$ are the parameters of the target networks, and $\hat{Q} = r + \gamma \min\{Q^{\bar{\omega}_1}(s', a'), Q^{\bar{\omega}_2}(s', a')\}$ with $a' \sim \pi^\theta(\cdot \mid s')$. For brevity, from now on, we denote $Q(s, a) = \min\{Q^{\omega_1}(s, a), Q^{\omega_2}(s, a)\}$.

**Handling action bounds.** In continuous-control tasks, actions are typically bounded (e.g., $a \in [-1, 1]^d$). Prior methods address the action bounds mainly by heuristics, e.g., using a truncated Gaussian for sampling noise (Dong et al., 2025; Ma et al., 2025; Jain et al., 2025). However, the truncation operations can break the probability path and lead to suboptimal behaviors. Instead, in this paper, we handle the action bounds in a principled way. We learn the flow in an unconstrained latent space $u \in \mathbb{R}^d$ and map latents to actions via $a = \tanh(u)$.

The soft policy improvement target in action space is the Boltzmann distribution $\pi_{\text{new}}(a \mid s) \propto \exp\left(\frac{1}{\lambda}Q(s, a)\right)$. Under the change of variables $a = \tanh(u_1)$, the corresponding unnormalized target density in latent space becomes

$$\tilde{\pi}_{\text{new}}(u_1 \mid s) \propto \exp\left(\frac{1}{\lambda}Q(s, \tanh(u_1))\right) \left|\det \frac{d\tanh(u_1)}{du_1}\right|$$

$$= \exp\left(\frac{1}{\lambda}Q(s, \tanh(u_1))\right) \prod_{j=1}^{d} \operatorname{sech}^2(u_{1,j}),$$

where $\operatorname{sech}$ denotes the hyperbolic secant function, and $u_{1,j}$ denotes the $j$-th component of $u_1$. This Jacobian factor is essential for enforcing the correct Boltzmann distribution in action space.

For the flow policy, we parameterize the latent-space velocity field as $v_t^\theta(u_t, s)$. To sample an action, we draw $u_0 \sim p_0$ and integrate the ODE $\frac{du_t}{dt} = v_t^\theta(u_t, s)$ in $t \in [0, 1]$. The action is $a = \tanh(u_1)$. We also denote this process by $a \sim \pi^\theta(\cdot \mid s)$.

Given $s$, $t$, and $u_t$, we estimate the noise posterior mean $\mu_{0|t}(u_t, s)$ with (15) and test function $\Phi_t = \operatorname{diag}(\Lambda)$, where $\Lambda \in \mathbb{R}^d$. By definition $q_{0|t}^*(u_0 \mid u_t, s) \propto p_0(u_0)\tilde{\pi}_{\text{new}}(u_1 \mid s)$. We have

$$\mu_{0|t}(u_t, s) = \mathbb{E}_{u_0 \sim q_{0|t}^*(\cdot|u_t,s)}\left[u_0 + \operatorname{diag}(\Lambda)s_{0|t}^*(u_0, u_t, s)\right].$$

Applying SNIS, we obtain

$$\hat{\mu}[u_0 \mid t, u_t, s; \Lambda] = \sum_{i=1}^{K} w^{(i)}\left(u_0^{(i)} + \operatorname{diag}(\Lambda)s_{0|t}^*\left(u_0^{(i)}, u_t, s\right)\right),$$

where $w^{(i)}$ denotes the normalized importance weight. Finally, we set $\bar{u}_0 = \hat{\mu}[u_0 \mid t, u_t, s; \hat{\Lambda}]$ and $\bar{u}_1 = \frac{u_t - \beta_t \bar{u}_0}{\alpha_t}$.

The RFM velocity target is given by

$$\hat{v}_t(u_t, s) = \dot{\alpha}_t \bar{u}_1 + \dot{\beta}_t \bar{u}_0.$$

The actor loss is then

$$\mathcal{L}_\pi(\theta) = \mathbb{E}_{t,u_t,s}\left[\left\|v_t^\theta(u_t, s) - \hat{v}_t(u_t, s)\right\|_2^2\right].$$

We adopt a policy-induced proposal to sample $u_t$. Given state $s$, we sample $u_1$ by integrating the ODE in $t \in [0, 1]$ with $u_0 \sim \mathcal{N}(0, I_d)$. Then $u_t = \alpha_t u_1 + \beta_t \epsilon$, where $\epsilon \sim \mathcal{N}(0, I_d)$. Additionally, following prior works (Ding et al., 2024; Dong et al., 2025), we generate $M$ action candidates during sampling, then select the one with the highest Q value. Additional details on the SNIS computation, along with pseudocode for our algorithm, are provided in Appendix E.

# 5. Experiments

## 5.1. Toy Example

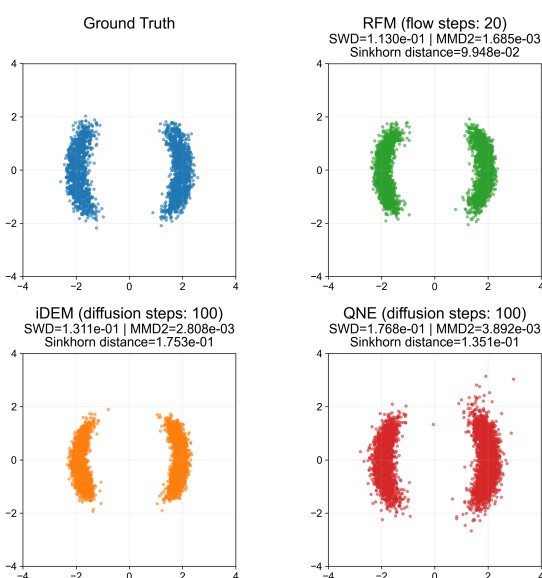

*Figure 1.* Two-moon target (top-left) and samples from three trained samplers (other panels). RFM (top-right) uses 20 steps to generate target samples, while the diffusion baselines iDEM (bottom-left) and QNE (bottom-right) use 100 steps. All methods are trained with the same posterior-estimation budget. Each algorithm panel reports sample-quality metrics against the ground-truth reference: sliced Wasserstein distance (SWD), squared maximum mean discrepancy (MMD$^2$), and Sinkhorn distance. RFM maintains the minimum discrepancy across all three metrics while requiring only one-fifth of the inference steps.

We first validate our RFM algorithm on a 2D two-moon target distribution, $p_1(x) \propto \exp(-E(x)/\lambda)$, where the density is known only up to a normalizing constant. During training, we only query the energy function $E(x)$ (and its

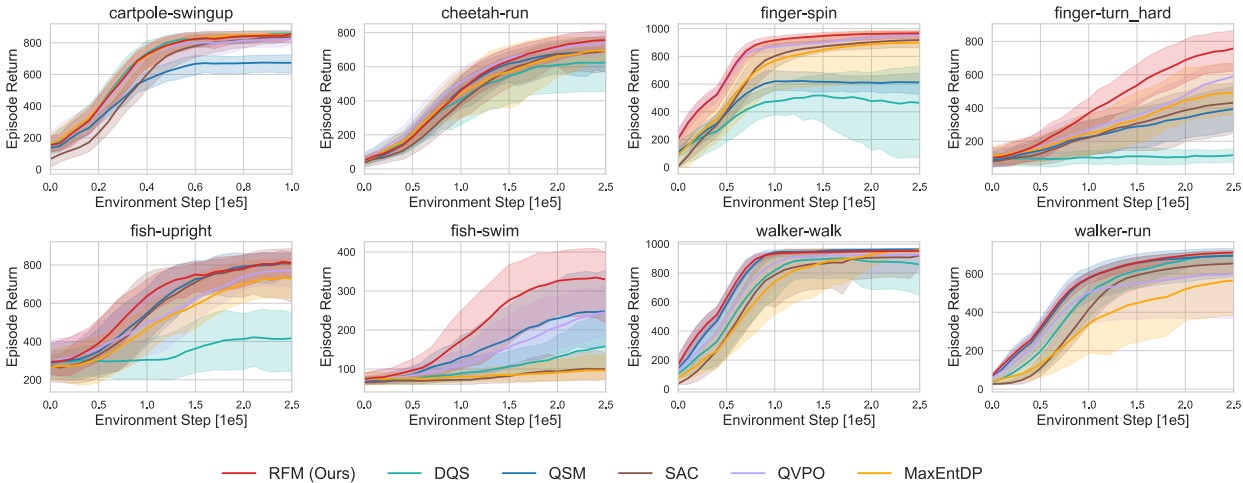

*Figure 2.* Training curves on eight environments. The solid lines correspond to the mean across five seeds, and the shaded regions represent minimum-to-maximum range over seeds. Our method (RFM) is the only algorithm that performs consistently well across all eight environments, exhibiting substantially better stability than the baselines.

gradient $\nabla E(x)$ when required), not target samples. We compare RFM's sample quality and inference-time efficiency against diffusion-based methods including iDEM (Akhound-Sadegh et al., 2024) (gradient-expectation) and QNE (Dong et al., 2025) (noise-expectation) under the same posterior-mean Monte Carlo sampling number, hence the same training computation. At inference time, RFM uses 20-step ODE integration, while the diffusion baselines use 100 denoising steps.

In Figure 1, RFM achieves the lowest distribution discrepancy across metrics while requiring only one-fifth of the inference steps compared to both iDEM and QNE. Experiment details are provided in Appendix F.

### 5.2. RL Tasks

We evaluate the proposed algorithm on eight environments from the DeepMind Control Suite (Tassa et al., 2018), a widely used continuous-control benchmark. We compare with the following baselines: (1) soft actor-critic (SAC) (Haarnoja et al., 2018), a standard maximum entropy RL method with Gaussian policies; (2) Q-score matching (QSM) (Psenka et al., 2024), which addresses Boltzmann sampling by training a score model to match the Q-function gradient and then sampling via Langevin dynamics; (3) MaxEntDP (Dong et al., 2025), whose core mechanism is Q-weighted noise estimation (QNE), a representative of the noise-expectation family for training diffusion policies to sample from Boltzmann distributions; (4) diffusion Q-sampling (DQS) (Jain et al., 2025), a representative of the gradient-expectation family for training diffusion policies to sample from Boltzmann distributions; and (5) Q-weighted variational policy optimization (QVPO) (Ding et al., 2024),

which adopts iterative weighted regression for policy improvement. Experiment details are reported in Appendix F.

The results are summarized in Figure 2. Our method (RFM) is the only algorithm that performs consistently well across all eight environments, and it exhibits substantially better stability than the baselines. In contrast, each baseline struggles severely on some of the tasks. Notably, RFM uses only 10 flow steps, whereas all other diffusion policy baselines use 20 diffusion steps. These results demonstrate the effectiveness of the reverse flow matching framework. By enabling the flow policy to learn a Boltzmann distribution, the advantages of flow models translate into improved performance, as reflected in both better total rewards and fewer inference steps. We report ablation studies, sensitivity analyses, and extended baseline comparisons in Appendix G.

## 6. Conclusion

In this paper, we proposed a unified framework for training diffusion and flow policies to sample from Boltzmann distributions in online RL. By adopting a reverse inferential perspective, we introduced reverse flow matching, which yields a tractable objective and turns the challenge of unavailable target samples into a posterior mean estimation problem. Moreover, we developed a general class of posterior mean estimators by leveraging Langevin Stein operators to construct control variates, reducing estimation variance and improving training stability. We also showed that existing methods for training diffusion policies to sample from Boltzmann distributions arise as special cases of our formulation. Empirically, our approach demonstrated stronger performance and greater stability across various continuous-control tasks compared to state-of-the-art baselines.

## Impact Statement

This paper introduces reverse flow matching (RFM), a unified framework for training diffusion and flow models to target unnormalized distributions without access to direct samples. Its goal is to advance the field of generative models and reinforcement learning (RL). RFM enables stable and efficient learning of expressive policies for continuous-control tasks, thereby supporting the deployment of RL methods in increasingly complex real-world applications.

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

## A. Related Works

In this section, we discuss prior works on leveraging diffusion and flow policies in online RL. We note that our review focuses on off-policy RL algorithms with diffusion and flow policies, primarily due to their sample efficiency and strong empirical performance. On-policy algorithms are also an active area of research, such as (Ding et al., 2026; Zhang et al., 2025). We categorize existing methods into four groups.

*Optimization via Differentiable Sampling.* This approach treats the sampling process of diffusion or flow models as a differentiable computational graph and directly optimizes the policy parameters by backpropagating the gradient of the Q-function through the chain. Wang et al. (2024) employ the reparameterization trick to differentiate through the diffusion sampling steps, enabling end-to-end policy optimization. Lv et al. (2025) backpropagate gradients through the deterministic flow sampling procedure to update the velocity field parameters. Celik et al. (2025) optimize a lower bound on the maximum entropy objective.

*Sampling via Langevin Dynamics.* Instead of learning a policy to directly output optimal actions, this class of methods leverages the Q-function as an energy landscape to guide action generation. Psenka et al. (2024) propose fitting a score network to the gradient of the Q-function, subsequently using Langevin dynamics to sample actions from the implied Boltzmann distribution. Ishfaq et al. (2025) integrate Langevin dynamics into the maximum entropy RL framework, using an uncertainty-aware critic to refine action sampling and improve exploration efficiency.

*Iterative Weighted Regression.* This paradigm simplifies the online RL problem by treating it as a repeated sequence of offline RL phases. States are sampled from the replay buffer, and the policy is updated on these states using a weighted form of the diffusion or flow matching loss. Ding et al. (2024) propose a Q-weighted diffusion policy loss to shift the policy towards high-Q actions. Fan et al. (2025) fine-tune flow models with online RL using a reward-weighted flow matching objective. Ma et al. (2025) introduce diffusion policy mirror descent (DPMD), which iteratively refines a diffusion policy via a Q-weighted diffusion loss. Gao et al. (2026) augment the weighted loss with entropy regularization.

*Targeting Boltzmann Distributions.* As discussed in the Introduction, this class of methods explicitly trains diffusion policies to sample from the Boltzmann distribution induced by the Q-function. The gradient-expectation family: Akhound-Sadegh et al. (2024) propose the iterated denoising energy matching (iDEM) algorithm, which constructs training targets via a weighted average of energy gradients. Jain et al. (2025) introduce diffusion Q-sampling (DQS), adapting iDEM to online RL by using a weighted average of Q-function gradients. The noise-expectation family: Ma et al. (2025) propose soft diffusion actor-critic (SDAC), which performs a weighted average over noise to construct the training target. Dong et al. (2025) introduce Q-weighted noise estimation (QNE), which similarly leverages a weighted noise average for training.

## B. Prior Methods as Special Cases

In this section, we explain in detail how several prior methods (Akhound-Sadegh et al., 2024; Jain et al., 2025; Dong et al., 2025; Ma et al., 2025) for training diffusion policies to sample from Boltzmann distributions can be recovered as special cases of our reverse flow matching framework. This viewpoint unifies two seemingly distinct estimator families: a *noise-expectation* family (Ma et al., 2025; Dong et al., 2025) that uses only evaluations of $Q$ and averages over noise, and a *gradient-expectation* family (Akhound-Sadegh et al., 2024; Jain et al., 2025) that instead computes and averages over $\nabla Q$. In our formulation, both arise from the same posterior mean estimation problem, differing only in the choice of test functions used to construct control variates. For simplicity and to align with the derivations in this paper, we drop the state variable $s$, use $x_1$ to denote data (i.e., the sample; in RL, this corresponds to the action $a$), use $x_0$ to denote noise, and let $Q(x_1)$ correspond to $Q(s, a)$. The target Boltzmann distribution is $p_1(x_1) \propto \exp\left(\frac{1}{\lambda}Q(x_1)\right)$.

Recall from Theorem 4.14 that for fixed $t$ and $x_t$, if we choose the isotropic test function $\Phi_t = \eta I_d$ (where $\eta \in \mathbb{R}$ is a constant) and assume a standard Gaussian source $p_0 = \mathcal{N}(0, I_d)$, the posterior mean estimator for $X_0$ is given by

$$\mu_{0|t}(x_t) = (1 - \eta)\, \mathbb{E}_{X_0 \sim q^*_{0|t}(\cdot|x_t)}[X_0] + \eta\, \mathbb{E}_{X_0 \sim q^*_{0|t}(\cdot|x_t)}\left[-\frac{1}{\lambda}\frac{\beta_t}{\alpha_t}\left[\nabla_{x_1}Q(x_1)\right]_{x_1 = \frac{1}{\alpha_t}x_t - \frac{\beta_t}{\alpha_t}X_0}\right].$$

Similarly, the posterior mean estimator for $X_1$ can be obtained by starting from the data-posterior form and deriving the corresponding control variate. The result is

$$\mu_{1|t}(x_t) = (1 - \eta)\mathbb{E}_{X_1 \sim q^*_{1|t}(\cdot|x_t)}[X_1] + \eta\left(\frac{1}{\alpha_t}x_t + \frac{1}{\lambda}\frac{\beta_t^2}{\alpha_t^2}\mathbb{E}_{X_1 \sim q^*_{1|t}(\cdot|x_t)}[\nabla Q(X_1)]\right).$$

Setting $\eta = 0$ yields

$$\begin{cases} \mu_{0|t}(x_t) = \mathbb{E}_{X_0 \sim q_{0|t}^*(\cdot|x_t)}[X_0], \\ \mu_{1|t}(x_t) = \mathbb{E}_{X_1 \sim q_{1|t}^*(\cdot|x_t)}[X_1]. \end{cases} \tag{22}$$

In contrast, setting $\eta = 1$ yields

$$\begin{cases} \mu_{0|t}(x_t) = \mathbb{E}_{X_0 \sim q_{0|t}^*(\cdot|x_t)}\left[ -\frac{1}{\lambda}\frac{\beta_t}{\alpha_t}\left[\nabla_{x_1} Q(x_1)\right]_{x_1 = \frac{1}{\alpha_t}x_t - \frac{\beta_t}{\alpha_t}X_0} \right], \\ \mu_{1|t}(x_t) = \frac{1}{\alpha_t}x_t + \frac{1}{\lambda}\frac{\beta_t^2}{\alpha_t^2}\mathbb{E}_{X_1 \sim q_{1|t}^*(\cdot|x_t)}[\nabla Q(X_1)]. \end{cases} \tag{23}$$

The iterated denoising energy matching (iDEM) algorithm proposed in (Akhound-Sadegh et al., 2024) uses the data-posterior estimator $\mu_{1|t}(x_t)$ in the $\eta = 1$ case (23) and trains the model to predict the score. Under the standard Gaussian source $p_0 = \mathcal{N}(0, I_d)$, the score function can be written as

$$s_t(x_t) = -\frac{1}{\beta_t}\mathbb{E}[X_0 \mid X_t = x_t] = -\frac{1}{\beta_t}\left(\frac{1}{\beta_t}x_t - \frac{\alpha_t}{\beta_t}\mathbb{E}[X_1 \mid X_t = x_t]\right) = -\frac{1}{\beta_t^2}x_t + \frac{\alpha_t}{\beta_t^2}\mathbb{E}[X_1 \mid X_t = x_t].$$

Replacing $\mathbb{E}[X_1 \mid X_t = x_t]$ with

$$\mu_{1|t}(x_t) = \frac{1}{\alpha_t}x_t + \frac{1}{\lambda}\frac{\beta_t^2}{\alpha_t^2}\mathbb{E}_{X_1 \sim q_{1|t}^*(\cdot|x_t)}[\nabla Q(X_1)],$$

we obtain

$$s_t(x_t) = -\frac{1}{\beta_t^2}x_t + \frac{\alpha_t}{\beta_t^2}\left(\frac{1}{\alpha_t}x_t + \frac{1}{\lambda}\frac{\beta_t^2}{\alpha_t^2}\mathbb{E}_{X_1 \sim q_{1|t}^*(\cdot|x_t)}[\nabla Q(X_1)]\right)$$
$$= \frac{1}{\lambda}\frac{1}{\alpha_t}\mathbb{E}_{X_1 \sim q_{1|t}^*(\cdot|x_t)}[\nabla Q(X_1)].$$

iDEM adopts the variance-exploding (VE) noise schedule in diffusion models. In our notation, this corresponds to setting $\alpha_t = 1$ and $\beta_t = \sigma_{\min}\left(\frac{\sigma_{\max}}{\sigma_{\min}}\right)^{1-t}$. Note that we use the forward-time convention, hence the exponent $1 - t$. Substituting $\alpha_t = 1$ into the expression above yields $s_t(x_t) = \frac{1}{\lambda}\mathbb{E}_{X_1 \sim q_{1|t}^*(\cdot|x_t)}[\nabla Q(X_1)]$. If we take $E = -Q$ and $\lambda = 1$, then $s_t(x_t) = -\mathbb{E}_{X_1 \sim q_{1|t}^*(\cdot|x_t)}[\nabla E(X_1)]$, which matches exactly the training target used by iDEM. Recall that $q_{1|t}^*(x_1 \mid x_t) \propto p_1(x_1)\, p_0\left(\frac{1}{\beta_t}x_t - \frac{\alpha_t}{\beta_t}x_1\right)$. For $p_0 = \mathcal{N}(0, I_d)$, the factor $p_0\left(\frac{1}{\beta_t}x_t - \frac{\alpha_t}{\beta_t}x_1\right)$, viewed as a function of $x_1$, is essentially a Gaussian $\mathcal{N}\left(\frac{1}{\alpha_t}x_t, \frac{\beta_t^2}{\alpha_t^2}I_d\right)$, which is exactly the proposal distribution for $x_1$ used by iDEM when performing SNIS. The diffusion Q-sampling (DQS) algorithm in (Jain et al., 2025) follows the iDEM approach. The training target of DQS is exactly $\frac{1}{\lambda}\mathbb{E}_{X_1 \sim q_{1|t}^*(\cdot|x_t)}[\nabla Q(X_1)]$.

The Q-weighted noise estimation (QNE) algorithm proposed in (Dong et al., 2025) uses the noise-posterior estimator $\mu_{0|t}(x_t)$ in the $\eta = 0$ case (22) and trains a noise-prediction network. The estimator is $\mathbb{E}_{X_0 \sim q_{0|t}^*(\cdot|x_t)}[X_0]$, which matches exactly the training target used by QNE. QNE uses the variance-preserving (VP) noise schedule in diffusion models, which corresponds to $\alpha_t = \exp\left(-\frac{1}{4}(1-t)^2(\beta_{\max} - \beta_{\min}) - \frac{1}{2}(1-t)\beta_{\min}\right)$ and $\beta_t = \sqrt{1 - \alpha_t^2}$. Note that we use the forward-time convention, and that $\beta_{\max}$ and $\beta_{\min}$ are a slight abuse of notation (unrelated to $\beta_t$). The proposal distribution for $x_0$ used by QNE when performing SNIS is $p_0 = \mathcal{N}(0, I_d)$, which is a natural choice since $q_{0|t}^*(x_0 \mid x_t) \propto p_0(x_0)\, p_1\left(\frac{1}{\alpha_t}x_t - \frac{\beta_t}{\alpha_t}x_0\right)$. The reweighted score matching (RSM) algorithm in (Ma et al., 2025) adopts the same estimator $\mu_{0|t}(x_t)$ as QNE and trains a score-prediction network. For $p_0 = \mathcal{N}(0, I_d)$, the score function is

$$s_t(x_t) = -\frac{1}{\beta_t}\mathbb{E}[X_0 \mid X_t = x_t] = -\frac{1}{\beta_t}\mu_{0|t}(x_t) = -\frac{1}{\beta_t}\mathbb{E}_{X_0 \sim q_{0|t}^*(\cdot|x_t)}[X_0],$$

which matches exactly the training target used by RSM. RSM uses a discrete-time diffusion model, i.e., a denoising diffusion probabilistic model (DDPM). It also uses $p_0 = \mathcal{N}(0, I_d)$ as the proposal distribution for $x_0$ for SNIS.

In summary, the noise-expectation family (QNE and RSM) corresponds to the noise-posterior estimator with $\eta = 0$, whereas the gradient-expectation family (iDEM and DQS) corresponds to the data-posterior estimator with $\eta = 1$. In both cases, these methods are recovered as concrete instantiations of our reverse flow matching framework, arising from particular choices of test functions used to construct control variates.

## C. Proofs of Theoretical Results

We restate each result before its proof for ease of reading.

**Proposition 4.1.** *The objectives $\mathcal{L}_{\text{RFM}}(\theta)$, $\mathcal{L}_{\text{RFM-N}}(\theta)$, and $\mathcal{L}_{\text{RFM-D}}(\theta)$ differ only by additive constants independent of $\theta$. Therefore, they share the same set of global minimizers and have identical gradients with respect to $\theta$.*

*Proof.* We focus on the noise-posterior objective $\mathcal{L}_{\text{RFM-N}}$, noting that the data-posterior case follows by symmetry. For brevity, write $V(X_0, X_1) = \dot{\alpha}_t X_1 + \dot{\beta}_t X_0$. We decompose $V(X_0, X_1)$ into its posterior mean and a residual term:

$$V(X_0, X_1) = \mathbb{E}[V(X_0, X_1) \mid X_t] + (V(X_0, X_1) - \mathbb{E}[V(X_0, X_1) \mid X_t]).$$

Consider the squared error loss conditioned on a fixed $t$ and $X_t$. Expanding the quadratic yields:

$$
\begin{aligned}
& \mathbb{E}_{(X_0, X_1) \sim q^*_{0,1|t}(\cdot | X_t)} \left[ \left\| v_t^\theta(X_t) - V(X_0, X_1) \right\|_2^2 \right] \\
&= \left\| v_t^\theta(X_t) - \mathbb{E}\left[V(X_0, X_1) \mid X_t\right] \right\|_2^2 + \mathbb{E}\left[ \left\| V(X_0, X_1) - \mathbb{E}\left[V(X_0, X_1) \mid X_t\right] \right\|_2^2 \mid X_t \right] \\
&\quad + 2 \left( v_t^\theta(X_t) - \mathbb{E}\left[V(X_0, X_1) \mid X_t\right] \right)^\top \mathbb{E}\left[V(X_0, X_1) - \mathbb{E}\left[V(X_0, X_1) \mid X_t\right] \mid X_t\right] \\
&= \left\| v_t^\theta(X_t) - \mathbb{E}\left[V(X_0, X_1) \mid X_t\right] \right\|_2^2 + \mathbb{E}\left[ \left\| V(X_0, X_1) - \mathbb{E}\left[V(X_0, X_1) \mid X_t\right] \right\|_2^2 \mid X_t \right],
\end{aligned}
\tag{24}
$$

where the last equality follows from the fact that the conditional expectation of the residual is zero, i.e., $\mathbb{E}[V(X_0, X_1) - \mathbb{E}[V(X_0, X_1) \mid X_t] \mid X_t] = 0$. The second term in the final expression represents the conditional variance of the target velocity, which is independent of $\theta$. Consequently, minimizing the original objective $\mathcal{L}_{\text{RFM}}$ is equivalent to minimizing the first term, which regresses $v_t^\theta(X_t)$ onto the posterior expectation of the velocity. To recover the specific form of $\mathcal{L}_{\text{RFM-N}}$, we express $X_1$ in terms of $X_t$ and $X_0$ via the interpolation constraint $X_1 = \frac{1}{\alpha_t} X_t - \frac{\beta_t}{\alpha_t} X_0$. Linearity of expectation yields:

$$
\begin{aligned}
\mathbb{E}[V(X_0, X_1) \mid X_t] &= \dot{\alpha}_t \mathbb{E}\left[ \frac{1}{\alpha_t} X_t - \frac{\beta_t}{\alpha_t} X_0 \;\middle|\; X_t \right] + \dot{\beta}_t \mathbb{E}[X_0 \mid X_t] \\
&= \frac{\dot{\alpha}_t}{\alpha_t} X_t + \left( \dot{\beta}_t - \frac{\dot{\alpha}_t \beta_t}{\alpha_t} \right) \mathbb{E}[X_0 \mid X_t].
\end{aligned}
$$

This matches the target defined in (9). Thus, $\mathcal{L}_{\text{RFM}}$ and $\mathcal{L}_{\text{RFM-N}}$ differ only by an additive constant, implying they share the same global minimizers and gradients. $\square$

**Theorem 4.2.** *Assume that for almost every $t \in [0, 1]$, the true marginal distribution $p_t$ (used in conditional flow matching) and the proposal distribution $\hat{p}_t$ (used in reverse flow matching) are mutually absolutely continuous. Assume that the parameterized function class $\{v_t^\theta : \theta \in \Theta\}$ is sufficiently rich such that the regression objectives attain their global minima. Then the objectives $\mathcal{L}_{\text{RFM-N}}$, $\mathcal{L}_{\text{RFM-D}}$, and $\mathcal{L}_{\text{CFM}}$ share the same set of global minimizers.*

*Proof.* By Proposition 4.1, $\mathcal{L}_{\text{RFM-N}}$ and $\mathcal{L}_{\text{RFM-D}}$ differ from $\mathcal{L}_{\text{RFM}}$ only by additive constants independent of $\theta$. Therefore, it suffices to show that $\mathcal{L}_{\text{RFM}}$ and $\mathcal{L}_{\text{CFM}}$ share the same set of global minimizers. Fix $t \in [0, 1]$ and write $V(X_0, X_1) = \dot{\alpha}_t X_1 + \dot{\beta}_t X_0$. In conditional flow matching, the forward construction samples $X_0 \sim p_0$ and $X_1 \sim p_1$, and then sets $X_t = \alpha_t X_1 + \beta_t X_0$. The conditional law of $(X_0, X_1)$ given $X_t$ induced by this construction coincides precisely with the posterior coupling $q^*_{0,1|t}(\cdot \mid X_t)$ used in reverse flow matching. For this fixed $t$, Lemma 3.1, combined with linear interpolation and (3), implies that $\mathcal{L}_{\text{CFM}}$ shares the same set of global minimizers as

$$\mathbb{E}_{X_t \sim p_t}\left[ \left\| v_t^\theta(X_t) - \mathbb{E}[V(X_0, X_1) \mid X_t] \right\|_2^2 \right], \tag{25}$$

where $p_t$ denotes the marginal distribution of $X_t$ under the forward construction. Analogously, the decomposition in (24) shows that the reverse flow matching loss is equivalent, up to an additive constant independent of $\theta$, to

$$\mathbb{E}_{X_t \sim \hat{p}_t}\left[ \left\| v_t^\theta(X_t) - \mathbb{E}[V(X_0, X_1) \mid X_t] \right\|_2^2 \right], \tag{26}$$

where $\hat{p}_t$ is the proposal used to sample $X_t$. Thus, the two objectives (25) and (26) share the same regression target $v_t(x) = \mathbb{E}[V(X_0, X_1) \mid X_t = x]$, differing only in the weighting measure over $X_t$. Under the stated richness assumption,

the set of global minimizers for (25) consists of those $v_t^\theta$ satisfying $v_t^\theta(x) = v_t(x)$ for $p_t$-almost every $x$, while the set for (26) consists of those satisfying the equality for $\hat{p}_t$-almost every $x$. Since $p_t$ and $\hat{p}_t$ are mutually absolutely continuous, these almost-everywhere conditions are equivalent, implying that the sets of global minimizers coincide. $\square$

**Lemma 4.5.** *Let $p$ be a continuously differentiable density on $\mathbb{R}^d$, and let $\phi : \mathbb{R}^d \to \mathbb{R}^d$ be continuously differentiable. Assume $\mathbb{E}_{X \sim p}[|(\mathcal{T}_p\phi)(X)|] < \infty$ and that*

$$\lim_{R \to \infty} \int_{\partial B_R} p(x)\,\phi(x) \cdot n(x)\,dS(x) \;=\; 0,$$

*where $B_R := \{x : \|x\|_2 \leq R\}$ and $n(x)$ denotes the outward unit normal on $\partial B_R$. Then*

$$\mathbb{E}_{X \sim p}[(\mathcal{T}_p\phi)(X)] \;=\; 0.$$

*Proof.* Since $p$ is $C^1$, we have $\nabla \log p = (\nabla p)/p$. Hence, by the product rule,

$$\nabla \cdot (\phi(x)p(x)) = p(x)\,\nabla \cdot \phi(x) + \phi(x) \cdot \nabla p(x) = p(x)\Big(\nabla \cdot \phi(x) + \phi(x) \cdot \nabla \log p(x)\Big) = p(x)\,(\mathcal{T}_p\phi)(x).$$

Integrating over $B_R$ and applying the divergence theorem yields

$$\int_{B_R} (\mathcal{T}_p\phi)(x)\,p(x)\,dx = \int_{B_R} \nabla \cdot (\phi(x)p(x))\,dx = \int_{\partial B_R} p(x)\,\phi(x) \cdot n(x)\,dS(x).$$

Taking $R \to \infty$ and using the assumed boundary condition gives $\int_{\mathbb{R}^d}(\mathcal{T}_p\phi)(x)\,p(x)\,dx \;=\; 0$, which is equivalent to $\mathbb{E}_{X \sim p}[(\mathcal{T}_p\phi)(X)] = 0$. $\square$

**Proposition 4.7.** *Assume that each column $\phi_j$ of $\Phi$ satisfies the integrability and boundary conditions of Lemma 4.5. Then*

$$\mathbb{E}_{X \sim p}[(\mathcal{T}_{p,m}\Phi)(X)] \;=\; 0 \in \mathbb{R}^m.$$

*Proof.* For each $j = 1, \cdots, m$, Lemma 4.5 implies $\mathbb{E}_{X \sim p}[(\mathcal{T}_p\phi_j)(X)] = 0$. Stacking these $m$ scalar identities yields $\mathbb{E}_{X \sim p}[(\mathcal{T}_{p,m}\Phi)(X)] = 0 \in \mathbb{R}^m$. $\square$

**Proposition 4.8.** *Let $\mu_{0|t}(x_t) = \mathbb{E}_{X_0 \sim q_{0|t}^*(\cdot|x_t)}[X_0]$. Assume $\Phi_t(\cdot, x_t)$ is admissible so that Proposition 4.7 applies. Then the estimator*

$$X_0 + (\mathcal{T}_{q_{0|t}^*(\cdot|x_t),d}\Phi_t(\cdot, x_t))(X_0)$$

*has zero variance under $q_{0|t}^*(\cdot \mid x_t)$ if and only if*

$$\Big(\mathcal{T}_{q_{0|t}^*(\cdot|x_t),d}\Phi_t(\cdot, x_t)\Big)(x_0) = \mu_{0|t}(x_t) - x_0 \tag{12}$$

*holds on the support of $q_{0|t}^*(\cdot \mid x_t)$.*

*Proof.* The estimator has zero variance if and only if it is constant on the support of $q_{0|t}^*(\cdot \mid x_t)$. Since its expectation is $\mu_{0|t}(x_t)$ by Proposition 4.7, the result (12) follows immediately. $\square$

Before presenting the proofs for our variance-reduction choices, we recall the asymptotic behavior of SNIS. This result provides an explicit objective for choosing $\Lambda$ to reduce the asymptotic variance.

**Lemma C.1.** *Let $q$ be a target distribution on $\mathbb{R}^d$ with unnormalized density $\tilde{q}$, and let $\bar{q}$ be a proposal distribution such that $q$ is absolutely continuous with respect to $\bar{q}$. Define the unnormalized importance weight*

$$w(x) := \frac{\tilde{q}(x)}{\bar{q}(x)}.$$

*Let $f : \mathbb{R}^d \to \mathbb{R}^d$ be measurable and define $\mu := \mathbb{E}_{X \sim q}[f(X)]$. Given samples $\{X^{(i)}\}_{i=1}^N \sim \bar{q}$, the self-normalized importance sampling estimator is*

$$\hat{\mu}_{\mathrm{SNIS}} := \frac{\sum_{i=1}^N w(X^{(i)})\,f(X^{(i)})}{\sum_{i=1}^N w(X^{(i)})}.$$

*Assume $\mathbb{E}_{\bar{q}}[w(X)] \in (0, \infty)$ and $\mathbb{E}_{\bar{q}}[w(X)^2 \|f(X)\|_2^2] < \infty$. Then, as $N \to \infty$,*

$$\sqrt{N}\left(\hat{\mu}_{\mathrm{SNIS}} - \mu\right) \Rightarrow \mathcal{N}(0, \Sigma),$$

*where the asymptotic covariance matrix is*

$$\Sigma = \frac{1}{\mathbb{E}_{\bar{q}}[w(X)]^2} \, \mathbb{E}_{\bar{q}}\left[w(X)^2 \left(f(X) - \mu\right)\left(f(X) - \mu\right)^\top\right]. \tag{27}$$

*Proof.* This is a classical result. See (Tokdar & Kass, 2010) for example. $\qquad\square$

Lemma C.1 shows that, for the $\Lambda$-controlled estimator (15), the asymptotic covariance is determined by the second moment of $w(x_0, x_t)^2\big(f_\Lambda(x_0, x_t) - \mu_{0|t}(x_t)\big)$ under the proposal $\bar{q}$, where

$$f_\Lambda(x_0, x_t) := x_0 + \mathrm{diag}(\Lambda)\, s_{0|t}^*(x_0, x_t).$$

We can therefore choose $\Lambda$ by minimizing the scalar criterion $\mathrm{tr}(\Sigma)$.

**Proposition 4.9.** *Fix $t$ and $x_t$. Let $X_0 \sim \bar{q}$ and let $w(X_0, x_t)$ denote the corresponding unnormalized importance weight. Recall the notations $s_{0|t,j}^*(x_0, x_t) := \partial_{x_{0,j}} \log q_{0|t}^*(x_0 \mid x_t)$ and $\mu_{0|t}(x_t) = \mathbb{E}_{X_0 \sim q_{0|t}^*(\cdot|x_t)}[X_0]$. Assume $\mathbb{E}_{\bar{q}}[w(X_0)] < \infty$ and $\mathbb{E}_{\bar{q}}[w(X_0)^2\|f_\Lambda(X_0, x_t)\|_2^2] < \infty$. Among all constant diagonal choices $h_{t,j} \equiv \Lambda_j$, the coefficients that minimize the asymptotic variance of estimator (15) are given component-wise by*

$$\Lambda_j^* = -\frac{\mathbb{E}_{\bar{q}}\left[w(X_0, x_t)^2\big(X_{0,j} - \mu_{0|t,j}(x_t)\big)s_{0|t,j}^*(X_0, x_t)\right]}{\mathbb{E}_{\bar{q}}\left[w(X_0, x_t)^2\big(s_{0|t,j}^*(X_0, x_t)\big)^2\right]}, \tag{16}$$

*for $j = 1, \cdots, d$.*

*Proof.* By Lemma C.1 with $f$ replaced by $f_\Lambda(\cdot, x_t)$, minimizing $\mathrm{tr}(\Sigma)$ is equivalent to minimizing

$$\mathbb{E}_{\bar{q}}\left[w(X_0, x_t)^2 \left\|f_\Lambda(X_0, x_t) - \mu_{0|t}(x_t)\right\|_2^2\right].$$

Using $f_\Lambda(x_0, x_t) = x_0 + \mathrm{diag}(\Lambda)s_{0|t}^*(x_0, x_t)$, the objective decomposes across coordinates:

$$\sum_{j=1}^d \mathbb{E}_{\bar{q}}\left[w(X_0, x_t)^2\left(X_{0,j} - \mu_{0|t,j}(x_t) + \Lambda_j s_{0|t,j}^*(X_0, x_t)\right)^2\right].$$

Each summand is a convex quadratic function of $\Lambda_j$. Differentiating with respect to $\Lambda_j$ and setting the derivative to zero yields (16). $\qquad\square$

**Proposition 4.10.** *Take the same assumptions as in Proposition 4.9. If further apply the isotropic restriction $\Lambda_1 = \cdots = \Lambda_d = \eta$ (equivalently, $\Phi_t = \eta I_d$), then the coefficient that minimizes the asymptotic variance of the corresponding SNIS estimator is*

$$\eta^* = -\frac{\mathbb{E}_{\bar{q}}\left[w(X_0, x_t)^2\big(X_0 - \mu_{0|t}(x_t)\big)^\top s_{0|t}^*(X_0, x_t)\right]}{\mathbb{E}_{\bar{q}}\left[w(X_0, x_t)^2\|s_{0|t}^*(X_0, x_t)\|_2^2\right]}. \tag{17}$$

*Proof.* By Lemma C.1 with $f$ replaced by $f_\eta(\cdot, x_t) = x_0 + \eta\, s_{0|t}^*(x_0, x_t)$, minimizing $\mathrm{tr}(\Sigma)$ is equivalent to minimizing

$$\mathbb{E}_{\bar{q}}\left[w(X_0, x_t)^2 \left\|f_\eta(X_0, x_t) - \mu_{0|t}(x_t)\right\|_2^2\right] = \mathbb{E}_{\bar{q}}\left[w(X_0, x_t)^2 \left\|\big(X_0 - \mu_{0|t}(x_t)\big) + \eta\, s_{0|t}^*(X_0, x_t)\right\|_2^2\right],$$

which is a convex quadratic function of $\eta$. Differentiating with respect to $\eta$ and setting the derivative to zero yields (17). $\quad\square$

**Theorem 4.14.** *Assume the target density has the Boltzmann form $p_1(x_1) \propto \exp\left(\frac{1}{\lambda}Q(x_1)\right)$. Let $\Phi_t(x_0, x_t) = \text{diag}\{h_{t,1}(x_0, x_t), \ldots, h_{t,d}(x_0, x_t)\}$ be a diagonal test function with constant entries $h_{t,j}(x_0, x_t) \equiv \Lambda_j$, and write $\Lambda = (\Lambda_1, \ldots, \Lambda_d)^\top$. Then the induced control variate satisfies*

$$g_{\Phi_t}(x_0, x_t) = \text{diag}(\Lambda) \nabla_{x_0} \log p_0(x_0) - \frac{1}{\lambda} \frac{\beta_t}{\alpha_t} \text{diag}(\Lambda) \left[\nabla_{x_1} Q(x_1)\right]_{x_1 = \frac{1}{\alpha_t} x_t - \frac{\beta_t}{\alpha_t} x_0}, \tag{18}$$

*and the posterior mean estimator can be expressed as*

$$\mu_{0|t}(x_t) = \mathbb{E}_{X_0 \sim q_{0|t}^*(\cdot|x_t)}\left[X_0 + \text{diag}(\Lambda) \nabla_{x_0} \log p_0(X_0) - \frac{1}{\lambda} \frac{\beta_t}{\alpha_t} \text{diag}(\Lambda) \left[\nabla_{x_1} Q(x_1)\right]_{x_1 = \frac{1}{\alpha_t} x_t - \frac{\beta_t}{\alpha_t} X_0}\right]. \tag{19}$$

*Moreover, if $p_0 = \mathcal{N}(0, I_d)$, we additionally have the identity*

$$\mathbb{E}_{X_0 \sim q_{0|t}^*(\cdot|x_t)}[X_0] = \mathbb{E}_{X_0 \sim q_{0|t}^*(\cdot|x_t)}\left[-\frac{1}{\lambda} \frac{\beta_t}{\alpha_t} \left[\nabla_{x_1} Q(x_1)\right]_{x_1 = \frac{1}{\alpha_t} x_t - \frac{\beta_t}{\alpha_t} X_0}\right]. \tag{20}$$

*If we further impose isotropic coefficients $\Lambda_j \equiv \eta$ for $j = 1, \ldots, d$ (equivalently, $\Phi_t = \eta I_d$), then the posterior mean simplifies to a linear combination:*

$$\mu_{0|t}(x_t) = (1 - \eta) \mathbb{E}_{X_0 \sim q_{0|t}^*(\cdot|x_t)}[X_0] + \eta \mathbb{E}_{X_0 \sim q_{0|t}^*(\cdot|x_t)}\left[-\frac{1}{\lambda} \frac{\beta_t}{\alpha_t} \left[\nabla_{x_1} Q(x_1)\right]_{x_1 = \frac{1}{\alpha_t} x_t - \frac{\beta_t}{\alpha_t} X_0}\right]. \tag{21}$$

*Proof.* Recall the control variate form $g_{\Phi_t}(x_0, x_t) = \text{diag}(\Lambda) \nabla_{x_0} \log q_{0|t}^*(x_0 \mid x_t)$. Using the posterior factorization $q_{0|t}^*(x_0 \mid x_t) \propto p_0(x_0) p_1\left(\frac{1}{\alpha_t} x_t - \frac{\beta_t}{\alpha_t} x_0\right)$, we have $\log q_{0|t}^*(x_0 \mid x_t) = \log p_0(x_0) + \frac{1}{\lambda} Q\left(\frac{1}{\alpha_t} x_t - \frac{\beta_t}{\alpha_t} x_0\right) + C$, where $C$ depends only on $x_t$. Differentiating with respect to $x_0$ via the chain rule, we have

$$\nabla_{x_0} \log q_{0|t}^*(x_0 \mid x_t) = \nabla_{x_0} \log p_0(x_0) + \frac{1}{\lambda} \left[\nabla_{x_1} Q(x_1)\right]_{x_1 = \frac{1}{\alpha_t} x_t - \frac{\beta_t}{\alpha_t} x_0} \cdot \left(-\frac{\beta_t}{\alpha_t} I_d\right).$$

Multiplying by $\text{diag}(\Lambda)$ produces (18). Substituting it into the identity $\mathbb{E}[X_0] = \mathbb{E}[X_0 + g_{\Phi_t}(X_0, x_t)]$ yields (19). For the case $p_0 = \mathcal{N}(0, I_d)$, we have $\nabla_{x_0} \log p_0(x_0) = -x_0$. Since $\mathbb{E}_{X_0 \sim q_{0|t}^*}[g_{\Phi_t}(X_0, x_t)] = 0$ holds for any $\Lambda$, we choose $\Lambda_j = 1$ to obtain $\mathbb{E}\left[-X_0 - \frac{1}{\lambda} \frac{\beta_t}{\alpha_t} \nabla_{x_1} Q(x_1)\right] = 0$, which proves (20). Finally, substituting $\Phi_t = \eta I_d$ and $\nabla_{x_0} \log p_0(x_0) = -x_0$ into the estimator $X_0 + g_{\Phi_t}$ yields (21). $\square$

## D. Reverse Score Matching

In this section, we demonstrate how to extend the reverse flow matching framework to learn score-based models when direct samples from the target distribution $p_1$ are unavailable and the source distribution $p_0$ goes beyond the Gaussian case. We name this extension *reverse score matching*.

Consider the stochastic differential equation (SDE) given by

$$dX_t = \left(v_t(X_t) + \frac{1}{2} \sigma_t^2 s_t(X_t)\right) dt + \sigma_t dW_t, \quad X_{t=0} \sim p_0,$$

where $\sigma_t > 0$ determines the diffusion level and $W_t$ is a standard Brownian motion. To simulate this process, one requires access to both the velocity field $v_t$ and the score function $s_t$.

The velocity field $v_t$ and the score function $s_t$ are intrinsic properties of the marginal probability path $(p_t)_{t \in [0,1]}$. However, the computational relationship between them depends crucially on the choice of source distribution $p_0$. When the source distribution is Gaussian (e.g., $p_0 = \mathcal{N}(0, I_d)$), the score function admits a closed-form expression and is linearly related to the velocity field. Consider the linear interpolation $X_t = \alpha_t X_1 + \beta_t X_0$. The conditional probability path $p_{t|1}(x_t \mid x_1)$ is Gaussian: $p_{t|1}(x_t \mid x_1) = \mathcal{N}(\alpha_t x_1, \beta_t^2 I_d)$. Consequently, the score function can be written as

$$s_t(x_t) = \nabla_{x_t} \log p_t(x_t) = -\frac{1}{\beta_t} \mathbb{E}[X_0 \mid X_t = x_t] = \frac{1}{\beta_t} \frac{\dot{\alpha}_t x_t - \alpha_t v_t(x_t)}{\alpha_t \dot{\beta}_t - \dot{\alpha}_t \beta_t}.$$

In this regime, learning the velocity field $v_t$ via reverse flow matching is sufficient to recover $s_t$ and enable SDE-based sampling.

For a general source distribution $p_0$, no such simple algebraic link exists. While $v_t$ and $s_t$ are mathematically coupled through the continuity equation and the Fokker–Planck equation, this connection does not, in general, yield a tractable expression for $s_t$ in terms of $v_t$. Consequently, even after learning $v_t$ via reverse flow matching, one must learn $s_t$ separately. We extend the reverse flow matching framework to reverse score matching, which allows us to train score-based models even when direct samples from $p_1$ are unavailable and the source distribution $p_0$ is arbitrary.

To learn the parameterized score function $s_t^\theta$, we begin with the standard (conceptual) score matching loss:

$$\mathcal{L}_{\mathrm{SM}}(\theta) = \mathbb{E}_{t \sim \mathcal{U}[0,1], X_t \sim p_t} \left[ \left\| s_t^\theta\left(X_t\right) - s_t\left(X_t\right) \right\|_2^2 \right].$$

As with flow matching, the marginal score $s_t(x)$ is intractable. We therefore resort to conditional score matching. The conditional score functions are defined as $s_{t|0}(x_t \mid x_0) = \nabla_{x_t} \log p_{t|0}(x_t \mid x_0)$ and $s_{t|1}(x_t \mid x_1) = \nabla_{x_t} \log p_{t|1}(x_t \mid x_1)$. The marginal and conditional scores are related via

$$s_t(x) = \mathbb{E}\left[ s_{t|1}\left(X_t \mid X_1\right) \mid X_t = x \right] = \mathbb{E}\left[ s_{t|0}\left(X_t \mid X_0\right) \mid X_t = x \right].$$

The conditional score matching objective regresses $s_t^\theta$ onto these conditional targets:

$$\begin{aligned}
\mathcal{L}_{\mathrm{CSM}}(\theta) &= \mathbb{E}_{t \sim \mathcal{U}[0,1], X_0 \sim p_0, X_t \sim p_{t|0}} \left[ \left\| s_t^\theta\left(X_t\right) - s_{t|0}\left(X_t \mid X_0\right) \right\|_2^2 \right] \\
&= \mathbb{E}_{t \sim \mathcal{U}[0,1], X_1 \sim p_1, X_t \sim p_{t|1}} \left[ \left\| s_t^\theta\left(X_t\right) - s_{t|1}\left(X_t \mid X_1\right) \right\|_2^2 \right] + \text{constant}.
\end{aligned}$$

Similar to Lemma 3.1, it is well-established that $\mathcal{L}_{\mathrm{SM}}$ and $\mathcal{L}_{\mathrm{CSM}}$ are equivalent up to a constant independent of $\theta$.

Conditional score matching relies on the forward construction of samples $(X_0, X_1)$ and interpolant $X_t$. To address the setting where $X_1 \sim p_1$ is unavailable, we apply the same reverse inference logic in reverse flow matching. Under linear interpolation $X_t = \alpha_t X_1 + \beta_t X_0$ and independent coupling, the reverse score matching loss is defined as

$$\begin{aligned}
\mathcal{L}_{\mathrm{RSM}}(\theta) &= \mathbb{E}_{t \sim \mathcal{U}[0,1], X_t \sim \hat{p}_t, X_0 \sim q_{0|t}^*(X_0 | X_t)} \left[ \left\| s_t^\theta\left(X_t\right) - s_{t|0}\left(X_t \mid X_0\right) \right\|_2^2 \right] \\
&= \mathbb{E}_{t \sim \mathcal{U}[0,1], X_t \sim \hat{p}_t, X_1 \sim q_{1|t}^*(X_1 | X_t)} \left[ \left\| s_t^\theta\left(X_t\right) - s_{t|1}\left(X_t \mid X_1\right) \right\|_2^2 \right] + \text{constant},
\end{aligned}$$

where $\hat{p}_t$ is a proposal distribution for $X_t$. Analogous to Proposition 4.1, we can derive noise-posterior and data-posterior variants by pushing the expectations inside the norm:

$$\mathcal{L}_{\mathrm{RSM-N}}(\theta) = \mathbb{E}_{t \sim \mathcal{U}[0,1], X_t \sim \hat{p}_t} \left[ \left\| s_t^\theta\left(X_t\right) - \mathbb{E}_{X_0 \sim q_{0|t}^*(X_0 | X_t)} \left[ \frac{1}{\beta_t} \nabla_{x_0} \log p_0\left(x_0\right) \right] \right\|_2^2 \right], \tag{28}$$

$$\mathcal{L}_{\mathrm{RSM-D}}(\theta) = \mathbb{E}_{t \sim \mathcal{U}[0,1], X_t \sim \hat{p}_t} \left[ \left\| s_t^\theta\left(X_t\right) - \mathbb{E}_{X_1 \sim q_{1|t}^*(X_1 | X_t)} \left[ \frac{1}{\alpha_t} \nabla_{x_1} \log p_1\left(x_1\right) \right] \right\|_2^2 \right]. \tag{29}$$

Note that, under linear interpolation and independent coupling, the conditional scores satisfy $s_{t|1}\left(x_t \mid x_1\right) = \frac{1}{\beta_t} s_0\left(\frac{x_t - \alpha_t x_1}{\beta_t}\right)$ and $s_{t|0}\left(x_t \mid x_0\right) = \frac{1}{\alpha_t} s_1\left(\frac{x_t - \beta_t x_0}{\alpha_t}\right)$, where $s_0 = \nabla_{x_0} \log p_0(x_0)$ and $s_1 = \nabla_{x_1} \log p_1(x_1)$ are the source and target scores, respectively. Additionally,

$$\mathbb{E}_{X_0 \sim q_{0|t}^*(X_0 | X_t)} [f(X_0)] = \mathbb{E}_{X_1 \sim q_{1|t}^*(X_1 | X_t)} \left[ f\left( \frac{1}{\beta_t} X_t - \frac{\alpha_t}{\beta_t} X_1 \right) \right].$$

The equivalence between the derived objectives is stated in the following propositions.

**Proposition D.1.** *The objectives $\mathcal{L}_{\mathrm{RSM}}(\theta), \mathcal{L}_{\mathrm{RSM-N}}(\theta)$ and $\mathcal{L}_{\mathrm{RSM-D}}(\theta)$ differ only by additive constants independent of $\theta$. Consequently, they share the same set of global minimizers and have identical gradients with respect to $\theta$.*

*Proof.* The proof mirrors that of Proposition 4.1. For a fixed $t$ and $X_t$, the target in $\mathcal{L}_{\mathrm{RSM}}$ is a random variable $S = s_{t|1}(X_t \mid X_1)$ distributed according to the posterior coupling. The quadratic loss decomposes into the squared error relative to the posterior mean $\mathbb{E}[S \mid X_t]$ and the posterior variance $\mathrm{Var}(S \mid X_t)$. Since the variance term is independent of $\theta$, minimizing $\mathcal{L}_{\mathrm{RSM}}$ is equivalent to regressing $s_t^\theta(X_t)$ onto $\mathbb{E}[S \mid X_t]$. By the specific forms of the conditional scores derived above, $\mathbb{E}[S \mid X_t]$ corresponds exactly to the targets in $\mathcal{L}_{\mathrm{RSM-N}}$. A similar argument applies to $\mathcal{L}_{\mathrm{RSM-D}}$. $\qquad\square$

**Proposition D.2.** *Under the assumptions of Theorem 4.2, the objectives $\mathcal{L}_{\mathrm{RSM-N}}(\theta)$, $\mathcal{L}_{\mathrm{RSM-D}}(\theta)$, and $\mathcal{L}_{\mathrm{CSM}}(\theta)$ share the same set of global minimizers.*

*Proof.* The proof mirrors that of Theorem 4.2. Both objectives regress $s_t^\theta(x)$ onto the marginal score $s_t(x)$. $\mathcal{L}_{\mathrm{CSM}}$ weighs the regression errors by the marginal density $p_t(x)$, while $\mathcal{L}_{\mathrm{RSM}}$ weighs them by the proposal density $\hat{p}_t(x)$. Since $p_t$ and $\hat{p}_t$ are mutually absolutely continuous, the sets of global minimizers coincide. $\qquad\square$

The variance reduction techniques proposed in this paper, such as Langevin Stein operators and control variates, are directly applicable to the expectations in (28) and (29), ensuring efficient training of the score function.

# E. Algorithm Details

We provide details for SNIS computation. The score of noise posterior is given by

$$s_{0|t}^* (u_0, u_t, s) = \nabla_{u_0} \log p_0 (u_0) - \frac{1}{\lambda} \frac{\beta_t}{\alpha_t} \nabla_{u_1} Q (s, \tanh (u_1)) + \frac{2\beta_t}{\alpha_t} \tanh (u_1),$$

where $u_1 = \frac{u_t - \beta_t u_0}{\alpha_t}$. Here $\nabla_{u_1} Q(s, \tanh(u_1))$ denotes the gradient with respect to the latent variable $u_1$, so the derivative of the $\tanh$ map is included. We use a standard Gaussian source $p_0 = \mathcal{N}(0, I_d)$, so $\nabla_{u_0} \log p_0(u_0) = -u_0$. We have

$$\mu_{0|t} (u_t, s) = \mathbb{E}_{u_0 \sim q_{0|t}^*(\cdot | u_t, s)} \left[ u_0 + \mathrm{diag}(\Lambda) s_{0|t}^* (u_0, u_t, s) \right].$$

To estimate $\mu_{0|t}(u_t, s)$, we use SNIS with control variates as in (15). Specifically, we draw $K$ samples $\{u_0^{(i)}\}_{i=1}^K \sim p_0$, and set $u_1^{(i)} = \frac{u_t - \beta_t u_0^{(i)}}{\alpha_t}$. We form the importance weights $\tilde{w}^{(i)} = \exp\left( \frac{1}{\lambda} Q\left( s, \tanh\left( u_1^{(i)} \right) \right) \right) \prod_{j=1}^d \mathrm{sech}^2\left( u_{1,j}^{(i)} \right)$ and the normalized weights $w^{(i)} = \frac{\tilde{w}^{(i)}}{\sum_{j=1}^K \tilde{w}^{(j)}}$. Then

$$\hat{\mu}[u_0 \mid t, u_t, s; \Lambda] = \sum_{i=1}^K w^{(i)} \left( u_0^{(i)} + \mathrm{diag}(\Lambda) s_{0|t}^* \left( u_0^{(i)}, u_t, s \right) \right).$$

The vector $\Lambda$ can be estimated from the same samples using Proposition 4.9 and (16). Concretely, we compute the weighted means $\hat{u}_0 = \sum_{i=1}^K w^{(i)} u_0^{(i)}$ and $\hat{s} = \sum_{i=1}^K w^{(i)} s_{0|t}^*(u_0^{(i)}, u_t, s)$, then estimate the $j$-th element of $\Lambda$ as

$$\hat{\Lambda}_j = -\frac{\sum_{i=1}^K \left( w^{(i)} \right)^2 \left( u_{0,j}^{(i)} - \hat{u}_{0,j} \right) \left( s_j^{(i)} - \hat{s}_j \right)}{\sum_{i=1}^K \left( w^{(i)} \right)^2 \left( s_j^{(i)} - \hat{s}_j \right)^2 + r},$$

where $j = 1, \cdots, d$, $s^{(i)}$ is shorthand for $s_{0|t}^*(u_0^{(i)}, u_t, s)$, and $r > 0$ is a small ridge term.

The overall procedure is summarized in Algorithm 1. Note that we abuse notation slightly by using $\beta$ for the learning rate, which is unrelated to the schedule $\beta_t$.

# F. Experiment Details

## F.1. Toy Example

**Target distribution.** We consider a two-moon target distribution $p_1(x) \propto \exp(-E(x)/\lambda)$, where $x = [x_1, x_2]^\top \in \mathbb{R}^2$. The temperature $\lambda = 1$ and the energy function is defined as

$$E(x) = \frac{1}{2} \left( \frac{\|x\|_2 - 2}{0.2} \right)^2 - \log \left( \exp \left[ -\frac{1}{2} \left( \frac{x_1 - 2}{0.3} \right)^2 \right] + \exp \left[ -\frac{1}{2} \left( \frac{x_1 + 2}{0.3} \right)^2 \right] \right).$$

---

**Algorithm 1:** Online Reinforcement Learning with Reverse Flow Matching

---

**Input:** network parameters $\theta$, $\omega_1$, $\omega_2$; target network parameters $\bar{\omega}_1 \leftarrow \omega_1$, $\bar{\omega}_2 \leftarrow \omega_2$; temperature $\lambda$; learning rate $\beta$;
       target smoothing coefficient $\tau$; replay buffer $\mathcal{D} \leftarrow \varnothing$.

**for** *each iteration* **do**
    **for** *each environment step* **do**
        Sample action $a \sim \pi^\theta(\cdot \mid s)$ and execute it in the environment;
        Observe next state $s'$ and reward $r$;
        Store transition $\mathcal{D} \leftarrow \mathcal{D} \cup \{(s, a, r, s')\}$.
    **end**
    **for** *each gradient step* **do**
        Sample a mini-batch from $\mathcal{D}$;
        Update critics $\omega_i \leftarrow \omega_i - \beta\nabla_{\omega_i}\mathcal{L}_Q(\omega_i)$ for $i \in \{1, 2\}$;
        Update actor $\theta \leftarrow \theta - \beta\nabla_\theta\mathcal{L}_\pi(\theta)$;
        Update target networks $\bar{\omega}_i \leftarrow \tau\omega_i + (1 - \tau)\bar{\omega}_i$ for $i \in \{1, 2\}$.
    **end**
**end**

---

**RFM instantiation.** We adopt the standard linear schedule $\alpha_t = t$ and $\beta_t = 1 - t$ with $t \in [t_{\min}, 1]$ where $t_{\min} = 0.02$. Under this schedule, the conditional velocity is $v_{t|0,1}(x_t \mid x_0, x_1) = \dot{\alpha}_t x_1 + \dot{\beta}_t x_0 = x_1 - x_0$. Therefore, RFM trains a velocity network $v_t^\theta(x_t)$ by regressing onto a posterior mean estimate of $(x_1 - x_0)$, as described in (8).

**Learned proposal for noise and data posterior sampling.** To enable efficient SNIS for posterior mean estimation, we fit a diagonal-covariance Gaussian mixture proposal $\bar{p}_1$ to the unnormalized target $\tilde{p}_1$ by maximizing the variational objective

$$\mathbb{E}_{x \sim \bar{p}_1}\big[\log\tilde{p}_1(x) - \log\bar{p}_1(x)\big],$$

where $\log\tilde{p}_1(x) = -E(x)/\lambda$. Maximizing it is equivalent to minimizing the reverse KL divergence $\mathrm{KL}(\bar{p}_1 \parallel p_1)$. This leads to closed-form GMM proposal distributions $\bar{q}_{1|t}(x_1 \mid x_t) \propto \bar{p}_1(x_1)\, p_0\big((x_t - \alpha_t x_1)/\beta_t\big)$ and $\bar{q}_{0|t}(x_0 \mid x_t) \propto p_0(x_0)\,\bar{p}_1\big((x_t - \beta_t x_0)/\alpha_t\big)$. Using these proposal distributions for SNIS can help improve the effective sample size.

**Metrics.** We use three metrics to quantitatively evaluate the quality of samples generated by the learned flow and diffusion models. These metrics capture complementary notions of distributional discrepancy between generated and ground-truth samples.

- **Sliced Wasserstein distance (SWD).** We approximate the sliced Wasserstein distance by projecting samples onto 50 random one-dimensional directions and averaging the resulting one-dimensional Wasserstein distances. This provides an efficient proxy for Wasserstein discrepancies in the toy example.

- **Squared maximum mean discrepancy ($\text{MMD}^2$).** We report $\text{MMD}^2$ with an RBF kernel as a nonparametric measure of discrepancy between generated and ground-truth samples. To ensure comparability across methods and checkpoints, we fix the kernel bandwidth using the median heuristic computed once from a large set of ground-truth samples, and use the same bandwidth for all evaluations.

- **Sinkhorn distance.** We compute the entropically regularized optimal transport cost between the empirical distributions of generated and reference samples. We use the squared Euclidean cost with a fixed regularization coefficient $\varepsilon = 10^{-3}$, and evaluate the Sinkhorn distance using 2,000 samples from each distribution.

### F.2. RL Tasks

**Hyperparameters.** We follow the official open-source implementations of DQS, QSM, MaxEntDP, and QVPO. For SAC, we follow the CleanRL implementation (https://github.com/vwxyzjn/cleanrl). To ensure a fair comparison, we integrate all methods into a unified JAX codebase. The shared hyperparameters are summarized in Table 1. For RFM, we fix the temperature at $\lambda = 0.02$ across all environments and use 100 Monte Carlo samples for posterior mean estimation.

*Table 1.* Shared hyperparameters.

| Hyperparameter | RFM | DQS | MaxEntDP | QSM | QVPO | SAC |
|---|---|---|---|---|---|---|
| Batch size | 256 | 256 | 256 | 256 | 256 | 256 |
| Discount factor $\gamma$ | 0.99 | 0.99 | 0.99 | 0.99 | 0.99 | 0.99 |
| Target smoothing coefficient $\tau$ | 0.005 | 0.005 | 0.005 | 0.005 | 0.005 | 0.005 |
| Number of hidden layers | 2 | 2 | 2 | 2 | 2 | 2 |
| Number of hidden units | 256 | 256 | 256 | 256 | 256 | 256 |
| Actor learning rate | 3e-4 | 3e-4 | 3e-4 | 3e-4 | 3e-4 | 3e-4 |
| Critic learning rate | 1e-3 | 1e-3 | 1e-3 | 1e-3 | 1e-3 | 1e-3 |
| Replay buffer size | 2.5e5 | 2.5e5 | 2.5e5 | 2.5e5 | 2.5e5 | 2.5e5 |
| Diffusion/flow steps | 10 | 20 | 20 | 20 | 20 | N/A |
| Number of action candidates | 32 | N/A | 32 | N/A | 32 | N/A |

**Training Time.** All experiments were conducted on a desktop equipped with an NVIDIA RTX 5090 GPU and an Intel Core Ultra 9 285K CPU. The average training time on the `walker-run` environment is reported in Table 2.

*Table 2.* Comparison of training time.

| Algorithm | RFM | DQS | MaxEntDP | QSM | QVPO | SAC |
|---|---|---|---|---|---|---|
| Training time (minutes) | 15 | 33 | 30 | 10 | 24 | 8 |

## G. Additional Experiments

### G.1. Sensitivity Analyses

We analyze how different hyperparameter settings affect performance. The results are summarized in Figure 3. Our algorithm is robust to the choice of $K$ (number of Monte Carlo samples) and $N$ (number of flow steps). It is also insensitive to $M$ (number of action candidates during sampling), as long as $M$ is not too small. In the main experiments, we set $K = 100$, $M = 32$, and $N = 10$.

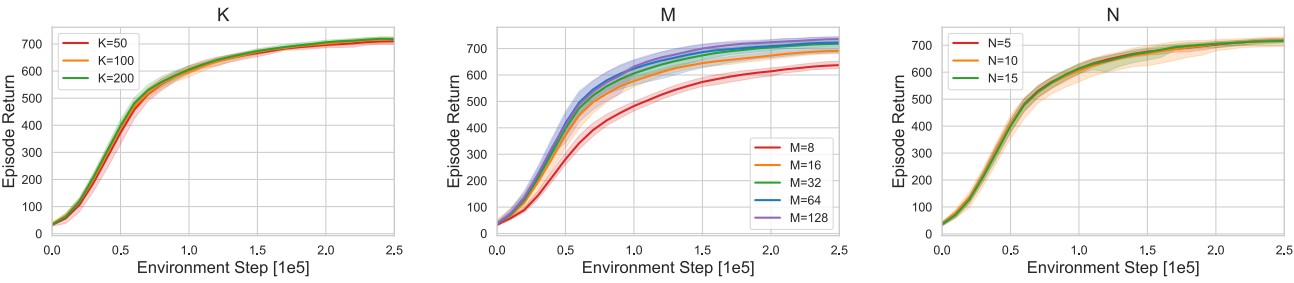

*Figure 3.* Training curves of RFM on `walker-run` environment under different hyperparameter settings. Left: experiments on different numbers of Monte Carlo samples $K$ for posterior mean estimation. Middle: experiments on different numbers of action candidates $M$ during sampling. Right: experiments on different flow steps during sampling.

### G.2. Ablation Studies

We analyze how different design choices affect performance on both the toy example and the RL tasks. First, for coefficients of control variates, our RFM algorithm estimates $\Lambda \in \mathbb{R}^d$ from samples. As an ablation, we fix the coefficients to be isotropic, setting $\Lambda_j \equiv \eta$ for $j = 1, \ldots, d$, where $\eta = 0$ recovers the noise-expectation form and $\eta = 1$ recovers the gradient-expectation form. Recall that $\mu_{0|t}(x_t) = (1 - \eta)\mathbb{E}[X_0] + \eta\,\mathbb{E}\left[-\frac{1}{\lambda}\frac{\beta_t}{\alpha_t}\nabla_{x_1}Q(x_1)\right]$. We compare these two fixed variants with RFM on the `finger-turn_hard` environment. As shown in Figure 4, the two fixed variants perform similarly to each other, but both are inferior to RFM. We observe the same pattern in the toy example (Figure 5): under the

same computational budget, RFM produces visibly and quantitatively better samples than the two fixed variants. Note that, in the toy example, all three methods can perform well with sufficient training; thus, the comparison is intended to highlight performance under limited computation.

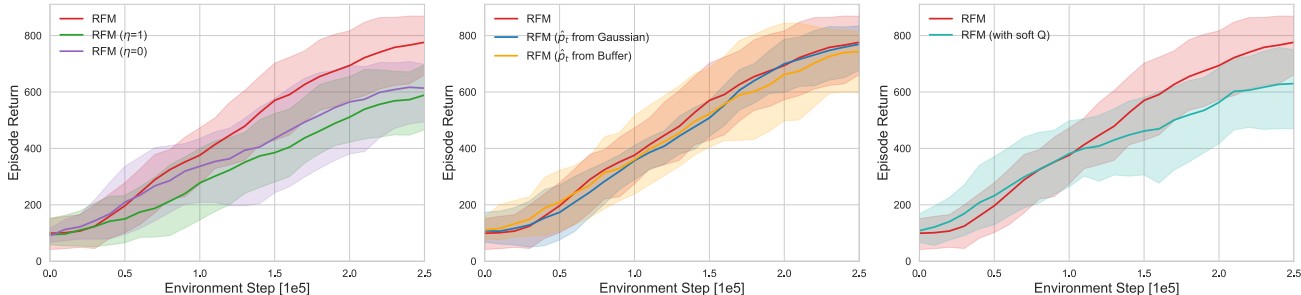

*Figure 4.* Training curves of RFM on `finger-turn_hard` environment under different design choices. Left: experiments on the effectiveness of control variates. Middle: experiments on choices of proposal distribution $\hat{p}_t$. Right: experiments on whether to use soft $Q$-function.

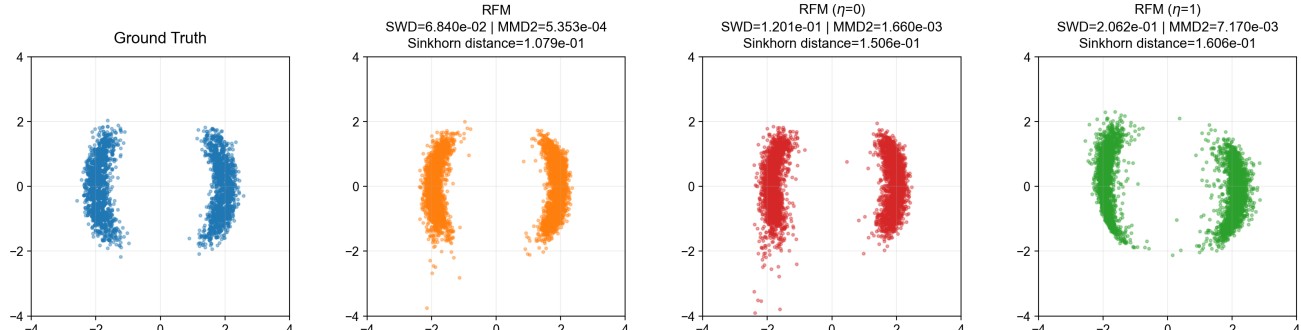

*Figure 5.* Experiments on the effectiveness of control variates in the toy example.

Second, we study the influence of proposal distribution $\hat{p}_t$. By Theorem 4.2, the key requirement on $p_t$ and $\hat{p}_t$ is mutual absolute continuity, namely that they have the same support. This assumption is satisfied in our experiments. In the toy example, the data domain is $\mathbb{R}^d$, so both $p_t$ and $\hat{p}_t$ have support on $\mathbb{R}^d$. In the RL experiments, we learn the flow in an unconstrained latent space $u \in \mathbb{R}^d$ and map latents to actions through $a = \tanh(u)$; consequently, both $p_t$ and $\hat{p}_t$ have support on $\mathbb{R}^d$ in the latent space. For the RL experiments, we evaluate three choices of $\hat{p}_t$ on the `finger-turn_hard` environment, as shown in Figure 4. The first choice, *from policy*, is the default in RFM: we sample $u_1$ from the current flow policy, sample $u_0$ from a standard Gaussian, and linearly interpolate them to obtain $u_t$. The second choice, *from buffer*, samples an action $a$ from the replay buffer, maps it to the latent space via $u_1 = \text{arctanh}(a)$, samples $u_0$ from a standard Gaussian, and then interpolates to obtain $u_t$. The third choice, *from Gaussian*, samples $u_t$ directly from a standard Gaussian. In the toy example, where no replay buffer is used, we compare the *from policy* and *from Gaussian* choices (Figure 6). Empirically, we observe no major performance difference among these choices in either setting. This is consistent with the theoretical result that the global optimizer does not depend on $\hat{p}_t$, and it suggests that training is robust to the choice of $\hat{p}_t$ in our tasks.

Third, we study the effect of using a soft $Q$-function. In RFM, we use the standard $Q$-function as the critic, rather than the soft $Q$-function commonly used in maximum-entropy RL (Haarnoja et al., 2018). This choice is mainly computational: for flow policies, evaluating the action log-density typically requires tracking the log-Jacobian along the ODE trajectory, for example via an augmented state, or performing an additional backward ODE integration, both of which increase training cost. To assess this design choice, we compare RFM with a variant that uses the soft $Q$-function during training. In this soft-$Q$ variant, we additionally incorporate the log-likelihood term and estimate it using Hutchinson's trace estimator. The training curves are shown in Figure 4. Empirically, RFM outperforms the soft-$Q$ variant, and incorporating the log-likelihood term slightly degrades performance. One possible explanation is that likelihood computation in ODE-based generative models introduces additional numerical difficulty and instability.

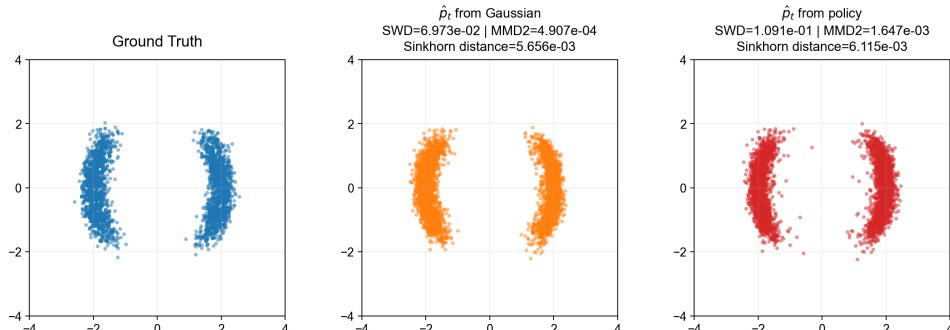

*Figure 6.* Experiments on choices of proposal distribution $\hat{p}_t$ in the toy example.

## G.3. Extended Comparisons

We further compare RFM with two additional baselines: (1) soft diffusion actor-critic (SDAC) (Ma et al., 2025), which belongs to the noise-expectation family for training diffusion policies to sample from Boltzmann distributions; and (2) diffusion-based maximum entropy RL (DIME) (Celik et al., 2025), which backpropagates through the sampling process to optimize a lower bound on the maximum-entropy objective. The results on the `finger-turn_hard` and `walker-run` environments are shown in Figure 7. Both baselines are outperformed by RFM. In particular, DIME exhibits unstable training, likely due to the long backpropagation chain through the sampling process.

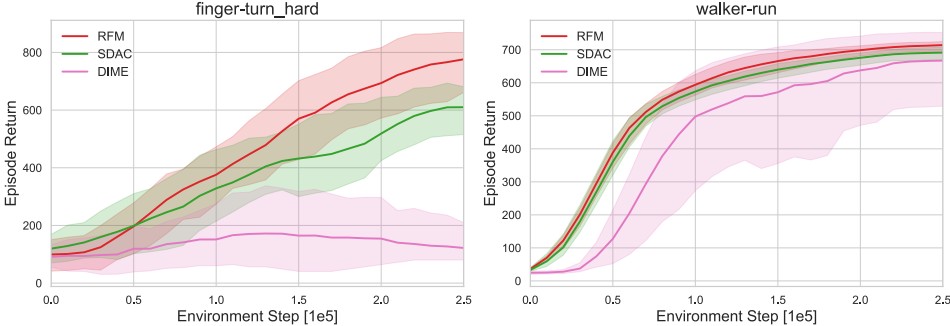

*Figure 7.* Training curves of RFM, SDAC, and DIME on `finger-turn_hard` and `walker-run` environments.

