# OpenReview forum: "Reverse Flow Matching: A Unified Framework for Online Reinforcement Learning with Diffusion and Flow Policies"
_ICML.cc/2026/Conference — ICML 2026 spotlight_

### Official Review · Reviewer_LCBJ · 2026-02-18

**Soundness:** 3
**Presentation:** 3
**Significance:** 3
**Originality:** 3
**Overall Recommendation:** 5
**Confidence:** 3

**Summary:**

The authors propose Reverse Flow Matching (RFM) that targets at improving diffusion / flow training for max entropy RL problems. In particular, the authors are motivated by a well-known issue that it is difficult to obtain direct action samples from Boltzmann distributions, and reformulate the problem using RFM. With the introduction of Langevin Stein operators, RFM can be viewed as a generalisation to previous noise / gradient expectation methods. The results on toy examples and standard DM Control Suite benchmarks have shown improved performance, including higher returns and efficiency.

**Compliance With Llm Reviewing Policy:**

Affirmed.

**Final Justification:**

As the main concerns are properly addressed, I've raised the score from 4 to 5.

**Key Questions For Authors:**

- Could the authors further explain / experiment on how the proposal distribution affects the optimisation? Also, in the toy example what is the proposal distribution?
- Could the authors perform ablation studies on the Langevin Stein operators, and see how it affects the performance?
- Could the authors clarify in appendix G.1, is $\bar{p}_1$ only used in RFM, or all comparators have benefited from the same / similar target proposal?

**Limitations:**

SNIS itself might now scale well with the dimensionality, which can be seen from the tasks evaluated where the action dimensionalities are all less than or equal to 6. It would be intereseting to see how it works with other tasks with higher number of action dimensions, e.g. humanoids.

**Strengths And Weaknesses:**

**strengths**
- The paper is well written. The motivation and the presentation are clear.
- The use of Langevin Stein operator elegantly stitches noise / gradient expectation methods together. This also provides a novel intepretation of previous methods in terms of control variates.

**weaknesses**
- The effect of the proposal distribution $\hat{p}(X_t)$ is not well discussed, while personally I think this is the most important contribution of this paper. I understand that Theorem 4.2 has stated that the choice of $\hat{p}(X_t)$  does not affect the minima, but as the authors acknowledged, the optimisation dynamics can be different. With certain choices of $\hat{p}(X_t)$ , the optimisation can be also quite difficult. In the RL setting (4.4), $u_t$ is obtained from samples of $u_0$ and $u_1$. In the toy example, however, this seems to be missing. Other choices of $\hat{p}(X_t)$ can be discussed. For example, what will happen if I just sample $X_t$ from simple $\hat{p}(X_t)$ like uniforms or gaussians?
- Another contribution of the paper is the use of Langevin Stein. However, the actual effect of this is not clearly ablated and tested.

---

> ### Author Rebuttal · Authors · 2026-03-31
>
> We sincerely thank the reviewer for their time and acknowledging the novelty and contributions of this work. We address the questions below.
>
> Q1: The effect of the proposal distribution $\hat{p}_t$ on learning performance.
>
> A1: Thank you for this insightful question. By Theorem 4.2, the key requirement on $p_t$ and $\hat{p}_t$ is mutual absolute continuity, i.e., that they have the same support. Under this condition, the two objectives share the same global optimizer, so in principle sufficient training should lead to the same solution regardless of the choice of $\hat{p}_t$. This assumption is satisfied in our experiments. In the toy example, the data domain is $\mathbb{R}^d$, so both $p_t$ and $\hat{p}_t$ assign nonzero probability on $\mathbb{R}^d$. In the RL experiments, we learn the flow in an unconstrained latent space $u\in\mathbb{R}^d$ and map latents to actions via $a=\tanh(u)$, so again both $p_t$ and $\hat{p}_t$ assign nonzero probability on $\mathbb{R}^d$.
>
> From a practical perspective, we agree that the choice of $\hat{p}_t$ can still affect optimization difficulty. We therefore study this effect in both the toy example and the RL setting. In the RL experiments, we consider three choices of $\hat{p}_t$ on the finger-turn_hard environment (Fig. 1, anonymous link): **from policy**, which samples $u_1$ from the current flow policy and $u_0$ from a standard Gaussian and then linearly interpolates to obtain $u_t$; **from buffer**, which samples $a$ from the replay buffer, maps it to the latent space by $u_1=\operatorname{arctanh}(a)$, samples $u_0$ from a standard Gaussian, and then interpolates to obtain $u_t$; and **from Gaussian**, which samples $u_t$ directly from a standard Gaussian. In the toy example, since no buffer is used, we compare **from policy** and **from Gaussian** (Fig. 2, anonymous link). Empirically, we observe no major performance difference across these choices in either setting. This is consistent with the theory that the global optimizer does not depend on $\hat{p}_t$, and it suggests that training is robust to the choice of $\hat{p}_t$ in our setting.
>
> Q2: Ablation studies of the control variates constructed from Langevin Stein operators.
>
> A2: Thank you for raising this point. We have conducted ablation studies of the control variates in both the toy example and the RL experiments. Specifically, we have compared RFM against two variants defined by $\\eta = 0$ and $\\eta = 1$ in
> $$
> \\mu_{0\\mid t}(x_t) = (1-\\eta)\\mathbb{E}[X_0] + \\eta\\,\\mathbb{E}\\left[-\\frac{1}{\\lambda}\\frac{\\beta_t}{\\alpha_t}\\nabla_{x_1}Q(x_1)\\right].
> $$
> Here, $\\eta = 0$ gives the **noise-expectation** form, while $\\eta = 1$ gives the **gradient-expectation** form.
>
> As shown in Fig. 3 (anonymous link), on finger-turn_hard, these two variants perform similarly, but both are inferior to RFM, which estimates the variance-minimizing coefficients from samples.
>
> We observe the same pattern in the toy example (Fig. 4, anonymous link): under the same computational budget, RFM yields visibly better sample quality than the two fixed variants, and the metrics confirm its superior performance. Of course, on this toy example, all three methods can perform well with sufficient training, and the comparison in Fig. 4 is intended to highlight performance under limited computation.
>
> Q3: Explanation of $\bar{p}_1$ in the toy example.
>
> A3: Thank you for raising this point. The GMM-fitted $\bar{p}_1$ is used only in RFM since the motivation for introducing it comes directly from our theoretical analysis, which reveals its relationship to the posterior distribution and $\bar{p}_1$.
>
> However, we have also incorporated $\bar{p}_1$ into the baselines and report the updated sampling quality results in Fig. 5 (anonymous link). RFM remains consistently better than iDEM and QNE with $\bar{p}_1$ across all metrics.
>
> Q4: Higher dimensions.
>
> A4: Thank you for this point. We agree that it would be interesting to explore higher-dimensional settings, such as humanoid control. We would like to note that while we chose SNIS for its simplicity and its seamless integration into the online RL loop, the core RFM framework is not inherently tied to SNIS: the reverse flow matching objective requires only posterior mean estimation, and the proposed control variates based on Langevin Stein operators can be combined with alternative estimators, including MCMC or SMC, as well as additional components such as a learned test function motivated by Proposition B.4. We will highlight this is an important direction for future work.

---

> > ### Author Rebuttal · Reviewer_LCBJ · 2026-03-31
> >
> > I appreciate the additional experiments. I've raised the score to 5.

---

> > > ### Author Response · Authors · 2026-04-01
> > >
> > > We sincerely thank the reviewer for confirming that their concerns have been addressed and for increasing their score. We will include the additional discussion and results in the revised paper.

---

### Official Review · Reviewer_QSqo · 2026-03-12

**Soundness:** 3
**Presentation:** 3
**Significance:** 3
**Originality:** 3
**Overall Recommendation:** 5
**Confidence:** 3

**Summary:**

This paper considers the important problem of effectively using generative policies (GPs) like flow and diffusion for online RL.  One of the major issues in this setting is that unlike traditional gaussian policies, it's very hard to backpropagate a SAC style gradient through the policy, due to the iterative nature of the computational graph of GPs. Naive backpropagation can lead to well-known issues like vanishing/exploding gradients and/or numerical instability.

One line of methods trains the GPs to sample actions from the Boltzmann distribution induced by the Q-function. Previous works in this line perform 1.  self-normalized importance sampling (SNIS) over the noise (utilizing exponentiated Q-values as weights) or 2. gradient of the Q-function itself. This limits them to diffusion policies as these derivations don't carry over directly to flow policies.

This paper shows that 1. and 2. are in fact subsumed by a more general reverse-flow matching (RFM) framework, which shows that training GPs to sample from the Q-energy landscape is equivalent to posterior mean estimation of the noise, conditioned on the interpolant being sampled from a user-specified proposal distribution.

Specifically, 1. and 2. arise as extreme cases of a family of estimators for the posterior mean (by setting \eta = 0 and 1 respectively). This work further shows that both choices are suboptimal with respect to the variance of this estimator, and the estimator with least variance is obtained via a certain choice of the control variate coefficient which lies between the two extremes.

From a practical standpoint, the RFM framework enables training flow policies to match the Q-energy landscape, which previous work targeting the Boltzmann distribution could not handle.  And the performance and stability of the resulting algorithm are also quite better than the previous line of work because the RFM framework enables explicitly setting a near optimal control variate coefficient.

**Compliance With Llm Reviewing Policy:**

Affirmed.

**Final Justification:**

The rebuttal addressed my concerns, so I am happy to maintain the positive score.

**Key Questions For Authors:**

1. I am not very up to date on continuous control online-RL benchmarks, but do recent works in this space use the same as used in this paper? If there exist newer or harder benchmarks, it could be useful to have a comparison on those, but I also believe the current results are good enough to support the main conceptual claims in the paper.

**Limitations:**

yes

**Strengths And Weaknesses:**

Strengths.
1. The paper addresses an important and timely problem, is well written and easy to follow.

2. The RFM framework is intriguing, and I believe it can be a significant conceptual advance in utilizing flow matching and diffusion policies for online RL.

Weaknesses.
1. While not a weakness as such, it could be useful to have an expanded discussion on when targeting the Boltzmann distribution is effective over other classes of methods for online RL with flow/diffusion policies such as sampling via Langevin dynamics or iterative weighted regression.

---

> ### Author Rebuttal · Authors · 2026-03-31
>
> We sincerely thank the reviewer for their time and their thoughtful review of our paper. We address the questions below.
>
> Q1: Discussion of when targeting the Boltzmann distribution is more effective than alternative approaches, such as sampling via Langevin dynamics or iterative weighted regression.
>
> A1: Thank you for this insightful question. We agree that the paper would benefit from a clearer discussion of when explicitly targeting the Boltzmann distribution is preferable to other classes of methods for online RL with generative policies. A key appeal of targeting the Boltzmann distribution is that it corresponds exactly to the policy improvement step in maximum-entropy RL, while enabling the learned policy to sample directly from this distribution. This naturally supports multimodality and exploration. Moreover, learning to sample from Boltzmann distributions is fundamentally important in many scientific domains beyond online RL, including physics, biology, and chemistry. Therefore, the methodology developed in this paper may also be useful in a broader range of applications.
>
> Compared with Langevin-based methods, our approach amortizes sampling into the policy itself, rather than relying on a separate energy-based sampling procedure at inference time, which can introduce additional computational overhead. Compared with iterative weighted regression methods, our approach targets the Boltzmann policy more directly, instead of formulating online RL as a sequence of offline RL problems and optimizing a data-weighted surrogate objective.
>
> At the same time, we agree that these alternatives may be preferable in certain regimes. Langevin-based methods can be attractive when iterative inference-time sampling is acceptable and one is willing to incur additional computation for improved performance. Iterative weighted regression methods are often conceptually simpler and can more readily leverage existing offline RL techniques in the online setting. We will add a paragraph to the paper to make this positioning more explicit.
>
> Q2: Continuous-control benchmarks for online RL.
>
> A2: Thank you for this important question. The DeepMind Control Suite used in this work remains a standard and widely used benchmark in the literature on online RL with generative policies. We agree that some recent works have also started to consider more challenging settings beyond these standard tasks. We therefore view our experiments as validating the main conceptual and theoretical claims on a widely used benchmark, and we will clarify in the paper that broader evaluation on more challenging settings is an important direction for future work.

---

> > ### Author Rebuttal · Reviewer_QSqo · 2026-04-03
> >
> > Thanks for the discussion regarding alternative approaches and benchmarks. I am happy to maintain my positive score.

---

> > > ### Author Response · Authors · 2026-04-03
> > >
> > > We sincerely appreciate your thoughtful reply and positive feedback.

---

### Official Review · Reviewer_B3Yg · 2026-03-13

**Soundness:** 3
**Presentation:** 3
**Significance:** 4
**Originality:** 4
**Overall Recommendation:** 5
**Confidence:** 4

**Summary:**

This paper utilizes a reformulation of flow matching, named reverse flow matching (RFM), to learn flow/diffusion-based policy in reinforcement learning. It first derives the posterior distributions (6) and (7) so that one can learn the vector field by reverse sampling (8). It then shows that it admits the same solution as the forward matching objective under certain condition (Thm.4.2). By utilizing the Langevin Stein operator and that the target distribution is a Boltzmann distribution, subsequent derivation reveals an self-normalized importance sampling (SNIS) estimate (14) that can be computed effectively, which then leads to efficient vector field training for action sampling of a policy. Experiments on both toy and DeepMind Control Suite show that RFM can outperform other flow-based algorithms.

**Compliance With Llm Reviewing Policy:**

Affirmed.

**Final Justification:**

The paper derives a theoretically motivated algorithm. The presentation is clear, and the empirical results are promising. The rebuttal sufficiently addressed my concerns, so I recommend accepting the paper and encourage the authors to include the clarifications in the revised version.

**Key Questions For Authors:**

Q1. Thm.4.2 assumes mutual abs continuous between $p_t$ and $\hat{p}_t$. Is this reasonable? Shouldn’t there be a restriction on the choice of $\eta$ in Thm.4.5?

Q2. What is the computational cost compared to other baselines?

**Limitations:**

Yes

**Strengths And Weaknesses:**

*Soundness:*

- Strengths: (1) One of the key ideas in this paper is to utilize the Boltzmann structure of the target policy (from policy improvement), combine it with SNIS and the Stein operator to derive a tractable vector field. It is technically sound and the derivation makes sense. The paper also nicely handles the Gaussian policy in the latent space, which is frequently ignored or heuristically addressed using truncation by other papers. (2) The proposed method is verified using both a toy example and common online RL problems to show that it is effective.
- Weaknesses: (1) Some of the theoretical results should be discussed further. For instance, Thm.4.2 assumes mutual abs continuous between $p_t$ and $\hat{p}_t$. Is this reasonable? Shouldn’t there be a restriction on the choice of $\eta$ in Thm.4.5? (2) It would be nice to have a comparison of the computational cost in the experiment. It would also be nice if we could see the effect of using the soft-Q value despite Remark 4.7.

*Presentation:*

- Strength: Despite being a dense paper with a lot of techniques involved, the exposition is clear and the flow is relatively easy to follow.
- Weakness: There is a gap that should be clarified in the paper. The RHS of (17) is a variance-reduced representation of the posterior mean. The SNIS estimate (11) is applied to the RHS (the inside of the expectation) of (17) instead of the original X_0 (L246 left). This is a bit convoluted as the SNIS was introduced in the beginning of Sec.4.2, so readers may assume that it will be used for $E[X_0]$ directly, but it is in fact used for the RHS of (17).

*Significance:* The paper addresses an important and relevant problem of efficiently training a Boltzmann policy, which provides both theoretical insight and practical impact.

*Originality:* The paper combines various techniques (Stein operator, SNIS, reverse objective) to derive a theoretically-motivated algorithm for training a flow-based policy, which is neat and clean.

---

> ### Author Rebuttal · Authors · 2026-03-31
>
> We sincerely thank the reviewer for their time and for acknowledging the novelty and contributions of this work. We address the questions below.
>
> Q1: Theorem 4.2 assumes $p_t$ and $\hat{p}_t$ are mutually absolutely continuous. Is this reasonable?
>
> A1: Thank you for this insightful question. The mutual absolute continuity condition is a rigorous way to state that $p_t$ and $\hat{p}_t$ have the same support, i.e., they assign nonzero probability to the same set. In Theorem 4.2, this assumption is needed to establish the equivalence of global minimizers under different weighting measures ($p_t$ and $\hat{p}_t$). We believe this is a reasonable assumption, and it is also satisfied in our experiments:
>
> In standard generative modeling tasks (e.g., the toy example), the data domain is $\mathbb{R}^d$, where $d$ is the data dimension, and hence both $p_t$ and $\hat{p}_t$ assign nonzero probability on $\mathbb{R}^d$. In our RL experiments, we learn the flow in an unconstrained latent space $u \in \mathbb{R}^d$, where $d$ is the action dimension, and then map the latent variable to the action space via $a=\tanh(u)$. Therefore, in this case as well, both $p_t$ and $\hat{p}_t$ assign nonzero probability on $\mathbb{R}^d$.
>
> Q2: In Theorem 4.6, should there be any restriction on $\eta$?
>
> A2: Thank you for raising this point. Eq. (19) is written as
>
> $$
> \mu_{0 \mid t}\left(x_t\right)=(1-\eta) \mathbb{E}\left[X_0\right]+\eta \mathbb{E}\left[-\frac{1}{\lambda} \frac{\beta_t}{\alpha_t}\nabla_{x_1} Q\left(x_1\right)\right]
> $$
>
> It is more convenient to rewrite it as
>
> $$
> \mu_{0 \mid t}\left(x_t\right)=\mathbb{E}\left[X_0\right]+\eta \mathbb{E}\left[-X_0-\frac{1}{\lambda} \frac{\beta_t}{\alpha_t}\nabla_{x_1} Q\left(x_1\right)\right]
> $$
>
> Under this form, the second term is a zero-mean control variate under the posterior, as established in Theorem 4.6. Hence, no restriction such as $\eta \in [0,1]$ is required for correctness. Any finite $\eta$ preserves the same expectation; it only affects the variance.
>
> Q3: Comparison of the computational cost in the experiment.
>
> A3: Thank you for this question. All experiments were conducted on a desktop equipped with an NVIDIA RTX 5090 GPU and an Intel Core Ultra 9 285K CPU. The average training time on the walker-run environment is reported below.
>
> | Algorithm | RFM | DQS | MaxEntDP | QSM | QVPO | SAC |
> |:---|:---:|:---:|:---:|:---:|:---:|:---:|
> | Training time (minutes) | 15 | 33 | 30 | 10 | 24 | 8 |
>
> Q4: The effect of using the soft-Q value.
>
> A4: Thank you for this question. We compare RFM with a soft-Q variant on the finger-turn_hard environment. In the soft-Q variant, we additionally incorporate the log-likelihood term and compute it using Hutchinson’s trace estimator [1]. The training curves are shown in this figure (anonymous link). Empirically, RFM outperforms the soft-Q variant, and incorporating the log-likelihood term slightly degrades performance. A possible reason is that likelihood computation in ODE-based generative models introduces additional numerical difficulty and instability. Prior work has shown that accurate likelihood estimation often depends on carefully designed objectives and problem-specific treatment [2].
>
> Q5: Clarification on which equation the SNIS is applied to.
>
> A5: Thank you for pointing this out. We agree that the current presentation may be somewhat confusing. In the revised paper, we will clarify exactly which equation SNIS is applied to and adjust the exposition to introduce SNIS at a more appropriate point.
>
> References:
>
> [1] Y. Lipman, M. Havasi, P. Holderrieth, N. Shaul, M. Le, B. Karrer, R. T. Q. Chen, D. Lopez-Paz, H. Ben-Hamu, and I. Gat. Flow Matching Guide and Code. arXiv:2412.06264, 2024.
>
> [2] K. Zheng, C. Lu, J. Chen, and J. Zhu. Improved Techniques for Maximum Likelihood Estimation for Diffusion ODEs. ICML, 2024.

---

> > ### Author Rebuttal · Reviewer_B3Yg · 2026-04-01
> >
> > I thank the authors for the reply and explanation. My concerns have been sufficiently addressed.
> >
> > The anonymous link in the rebuttal is missing and I encourage the authors to include relevant discussions in the appendix for completeness.

---

> > > ### Author Response · Authors · 2026-04-01
> > >
> > > We sincerely thank the reviewer for the positive follow-up and for confirming that their concerns have been fully addressed. We will incorporate the additional discussion and results into the revised paper.
> > >
> > > We apologize for the inconvenience regarding the anonymous links to the figures.
> > >
> > > The results for the soft-Q value ablation are available [here](https://anonymous.4open.science/r/rfm_rebuttal-BDEE/ablation_softQ_RL.pdf).
> > >
> > > For completeness, we also provide the figures corresponding to our responses to the other reviewers below.
> > >
> > > For Reviewer Y8DQ, Fig. 1 is available [here](https://anonymous.4open.science/r/rfm_rebuttal-BDEE/ablation_cv_RL.pdf). Fig. 2 is available [here](https://anonymous.4open.science/r/rfm_rebuttal-BDEE/ablation_cv_toy.png).
> > >
> > > For Reviewer LCBJ, Fig. 1 is available [here](https://anonymous.4open.science/r/rfm_rebuttal-BDEE/ablation_pt_RL.pdf). Fig. 2 is available [here](https://anonymous.4open.science/r/rfm_rebuttal-BDEE/ablation_pt_toy.png). Fig. 3 is available [here](https://anonymous.4open.science/r/rfm_rebuttal-BDEE/ablation_cv_RL.pdf). Fig. 4 is available [here](https://anonymous.4open.science/r/rfm_rebuttal-BDEE/ablation_cv_toy.png). Fig. 5 is available [here](https://anonymous.4open.science/r/rfm_rebuttal-BDEE/gmm_fit_comparison.png).

---

### Official Review · Reviewer_Y8DQ · 2026-03-13

**Soundness:** 4
**Presentation:** 3
**Significance:** 3
**Originality:** 4
**Overall Recommendation:** 5
**Confidence:** 4

**Summary:**

This paper introduces a unified framework for generative modeling of an unnormalized energy-based distribution with a known energy function but without access to samples from the distribution. Building upon the posterior mean estimation formulation, the paper proposes a unified framework, reverse flow matching (RFM), which unifies the previous noise-expectation and gradient-expectation methods as special cases. The authors also introduce Langevin Stein operators to derive estimators that share the same expectation but have lower variance. The performance of the proposed method is demonstrated on a toy generative modeling example and eight online RL environments.

**Compliance With Llm Reviewing Policy:**

Affirmed.

**Final Justification:**

My concerns have been adequately addressed.

**Key Questions For Authors:**

1. How does the control variate $g_{\Phi_t}(x_0, x_t)$ affect the policy learning performance in the online RL settings?
2. How does the proposed RFM RL algorithm perform compared to maximum-entropy diffusion policy baselines other than MaxEntDP such as SDAC [1] and DIME [2]?
3. Why in the final objectives the expectation over $X_0$ or $X_1$ is pushed into the squared norm? Can we achieve similar objectives with Langevin Stein operator based control variates but with this expectation outside the squared norm? In this way, the requirement for multiple Monte-Carlo samples can be avoided.

[1] Ma H, Chen T, Wang K, et al. Efficient Online Reinforcement Learning for Diffusion Policy[C]//International Conference on Machine Learning. PMLR, 2025: 41837-41853.

[2] Celik O, Li Z, Blessing D, et al. DIME: Diffusion-Based Maximum Entropy Reinforcement Learning[C]//International Conference on Machine Learning. PMLR, 2025: 6958-6977.

**Limitations:**

The limitations of this work should be discussed more thoroughly in the paper.

**Strengths And Weaknesses:**

Strengths

1. The proposed framework is novel and has significant impact on the field of diffusion/flow policies and generative modeling. The viewpoint of reverse inference and the formulation of posterior mean estimation under a given intermediate noisy sample is clear and elegant.
2. The paper is well-organized and easy to follow.
3. The claims in this paper are well-supported by rigorous derivations.

Weaknesses

1. The experiments lack comparison to some important online diffusion policy baselines in maximum-entropy RL, such as SDAC [1] and DIME [2]. There is also no ablation study on the generative modeling performance and the RL training performance w/ and w/o the Langevin Stein operator based control variate term $g_{\Phi_t}(x_0, x_t)$.
2. Minor issues. In Section 5.2, the mentioned baselines in the text does not align with the baselines shown in Figure 2. For example, QVPO [3] is not mentioned in the text, and [4] is denoted with 'QNE' in the text but 'MaxEntDP' in the Figure 2.

[1] Ma H, Chen T, Wang K, et al. Efficient Online Reinforcement Learning for Diffusion Policy[C]//International Conference on Machine Learning. PMLR, 2025: 41837-41853.

[2] Celik O, Li Z, Blessing D, et al. DIME: Diffusion-Based Maximum Entropy Reinforcement Learning[C]//International Conference on Machine Learning. PMLR, 2025: 6958-6977.

[3] Ding S, Hu K, Zhang Z, et al. Diffusion-based reinforcement learning via q-weighted variational policy optimization[J]. Advances in Neural Information Processing Systems, 2024, 37: 53945-53968.

[4] Dong X, Cheng J, Zhang X S. Maximum Entropy Reinforcement Learning with Diffusion Policy[C]//International Conference on Machine Learning. PMLR, 2025: 13963-13983.

---

> ### Author Rebuttal · Authors · 2026-03-31
>
> We sincerely thank the reviewer for their time and their thoughtful review of our paper. We address the questions below.
>
> Q1: How does the control variate affect policy learning performance?
>
> A1: Thank you for this question. We have conducted an ablation study on the control variate $g_{\\Phi_t}(x_0, x_t)$ in our RFM algorithm on the finger-turn_hard environment. Specifically, we have compared RFM against two variants defined by $\\eta = 0$ and $\\eta = 1$ in
>
> $$
> \\mu_{0\\mid t}(x_t) = (1-\\eta)\\mathbb{E}[X_0] + \\eta\\,\\mathbb{E}\\left[-\\frac{1}{\\lambda}\\frac{\\beta_t}{\\alpha_t}\\nabla_{x_1}Q(x_1)\\right].
> $$
>
> Here, $\\eta = 0$ gives the **noise-expectation** form, while $\\eta = 1$ gives the **gradient-expectation** form. As shown in Fig. 1 (anonymous link), these two variants perform similarly, but both are inferior to RFM, which estimates the variance-minimizing coefficients from samples.
>
> We have also conducted an ablation study on the toy example, and observe the same pattern (Fig. 2, anonymous link): under the same computational budget, RFM yields visibly better sample quality than the two fixed variants, and the metrics confirm its superior performance. Of course, on this toy example, all three methods can perform well with sufficient training, and the comparison in Fig. 2 is intended to highlight performance under limited computation.
>
> Q2: How does the proposed RFM algorithm compare with SDAC and DIME in terms of performance?
>
> A2: Thank you for pointing out these relevant baselines. SDAC is already cited in our paper. It belongs to the noise-expectation family, and we compared with the similar baseline in the same family, MatEntDP. We will make sure to cite DIME in the revised paper. We will also add additional comparative results including SDAC and DIME in the revised paper.
>
> Q3: In the final objective, why is the expectation over $X_0$ or $X_1$ taken inside the squared norm? Is it possible to define a similar objective with the expectation outside the squared norm, thereby avoiding the need for multiple Monte Carlo samples?
>
> A3: Thank you for this insightful question. We place the expectation inside the squared norm because the squared loss can be decomposed into a parameter-dependent term and a parameter-independent term. Let $Y_t=\dot{\alpha}_t X_1+\dot{\beta}_t X_0$. For fixed $X_t$, we have
>
> $$
> \mathbb{E}\left[\lVert v_\theta(X_t)-Y_t\rVert^2 \mid X_t\right] = \lVert v_\theta(X_t)-\mathbb{E}\left[Y_t \mid X_t\right]\rVert^2 + \operatorname{Var}\left(Y_t \mid X_t\right).
> $$
>
> The second term is independent of $\theta$. Therefore, Eq. (8) and Eqs. (9)-(10) have the same minimizers and gradients, as stated in Proposition 4.1.
>
> This simplification is a key benefit of reverse flow matching, which takes a reverse inferential perspective: we first sample $X_t$, and then sample $X_0$ from the posterior distribution. In contrast, standard conditional flow matching samples $X_0$ and $X_1$ first, and then constructs $X_t$ through linear interpolation. In that setting, moving the expectation inside the squared loss does not lead to a similarly meaningful simplification.
>
> We use the posterior-mean form because it makes the supervision target explicit and allows us to apply Langevin-Stein control variates directly to posterior mean estimation. Our construction yields zero-mean control variates under the posterior, and therefore preserves $\mathbb{E}[X_0 \mid X_t]$. In principle, one could keep the expectation outside the squared norm and still apply the control variate idea, but this would lead to a more unnecessarily complicated algorithm.
>
> Finally, the need for Monte Carlo samples does not come from placing the expectation inside the squared norm. Rather, it comes from the fact that the posterior is known only up to a normalization constant, so we cannot sample from it directly. Therefore, regardless of whether the expectation is placed inside or outside the squared norm, we still need to sample $X_0$ from a proposal distribution and use importance sampling.
>
> Q4: Description inconsistencies.
>
> A4: Thank you for pointing this out. QNE refers to the core mechanism underlying MaxEntDP. To avoid confusion and ensure consistency, we will revise the paper to use MaxEntDP uniformly as the algorithm name. We will also add a description of QVPO in the baseline paragraph.

---

> > ### Author Rebuttal · Reviewer_Y8DQ · 2026-04-03
> >
> > Thank you for the detailed reply. My concerns have been adequately addressed.

---

> > > ### Author Response · Authors · 2026-04-03
> > >
> > > Thank you sincerely for your positive follow-up and for confirming that your concerns have been fully addressed.

---

### Decision · Program_Chairs · 2026-04-30

**Decision:**

Accept (spotlight)

**Comment:**

The submission introduces Reverse Flow Matching (RFM), a unified framework for training expressive generative policies, such as diffusion and flow models, in online RL. The paper’s core contribution is a theoretical bridge that unifies two previously separate families of training objectives, i.e., noise-expectation and gradient-expectation methods, by reframing them as specific cases of a more general posterior mean estimation problem.

The reviewers were generally impressed by the theoretical soundness and strength of this work, and especially by its clarity. Reviewer LCBJ and Reviewer Y8DQ initially identified missing baseline comparisons (e.g., SDAC and DIME) and a lack of ablation on the Langevin-Stein operators, the authors’ thorough engagement during the rebuttal phase effectively addressed these gaps. Moreover, the rebuttal provided strong empirical evidence of the effectiveness of the proposed approach. The ablations on the finger-turn_hard environment has been helpful to demonstrate that RFM’s dynamic coefficient estimation outperforms both noise-expectation and gradient-expectation fixed variants. These results, along with the commitment to include additional state-of-the-art baselines like DIME, satisfied all reviewers.